# Comparative Analysis of Different Tube Models for Linear Rheology of Monodisperse Linear Entangled Polymers

**DOI:** 10.3390/polym11050754

**Published:** 2019-04-28

**Authors:** Volha Shchetnikava, Johan Slot, Evelyne van Ruymbeke

**Affiliations:** 1Department of Mathematics and Computer Science, Eindhoven University of Technology, 5600 MB Eindhoven, The Netherlands; j.j.m.slot@tue.nl; 2Dutch Polymer Institute (DPI), 5600 AX Eindhoven, The Netherlands; 3Bio and Soft Matter Group, Institute of Condensed Matter and Nanosciences, École Polytechnique de Louvain, Université catholique de Louvain, B-1348 Louvain-la-Neuve, Belgium; evelyne.vanruymbeke@uclouvain.be

**Keywords:** rheology, entangled polymer melts, tube model, linear chains

## Abstract

The aim of the present paper is to analyse the differences between tube-based models which are widely used for predicting the linear viscoelasticity of monodisperse linear polymers, in comparison to a large set of experimental data. The following models are examined: Milner–McLeish, Likhtman–McLeish, the Hierarchical model proposed by the group of Larson, the BoB model of Das and Read, and the TMA model proposed by the group of van Ruymbeke. This comparison allows us to highlight and discuss important questions related to the relaxation of entangled polymers, such as the importance of the contour-length fluctuations (CLF) process and how it affects the reptation mechanism, or the contribution of the constraint release (CR) process on the motion of the chains. In particular, it allows us to point out important approximations, inherent in some models, which result in an overestimation of the effect of CLF on the reptation time. On the contrary, by validating the TMA model against experimental data, we show that this effect is underestimated in TMA. Therefore, in order to obtain accurate predictions, a novel modification to the TMA model is proposed. Our current work is a continuation of earlier research (Shchetnikava et al., 2014), where a similar analysis is performed on well-defined star polymers.

## 1. Introduction

Currently, the theory of de Gennes, Doi and Edwards is considered to be the most advanced set of ideas in the physics of entangled polymers [1,2]. It replaces the complex system of linear polymer chains with a mean-field approach, employing the concept of a tube in order to represent the molecular environment of an entangled chain. Due to the simplifications it offers, the theory of de Gennes, Doi and Edwards, provides a powerful tool for applying rheology as an analytical method in industry, and the majority of research within the field of rheology follows this widely-accepted theoretical picture in order to describe the dynamics of macromolecules [3,4,5,6,7,8].

De Gennes [1] was the first to simplify the complex many-body problem of a large number of interacting polymer chains diffusing in a melt. His strategy was to introduce a mean-field approach by reducing the many-body problem (all chains) to a one-body problem (“test” chain) in an average effective field. The topological constraints imposed by the surrounding chains on the “test” chain are assumed to be equivalent to the tube, in which the “test” chain is confined since it cannot move in the direction perpendicular to the tube axis. At equilibrium, the tube is assumed to be a random Gaussian walk consisting of *Z* steps of length *a* and average length L=Za [2]. Nevertheless, when a rapid strain is imposed on the melt, the tube is deformed and the assumption of the random walk is not valid anymore. The following relaxation towards equilibrium occurs via a change in the configuration of the tube which confines the “test” chain. By wriggling back and forth, the chain vacates parts of the original deformed tube and the stress linked to those portions is “forgotten”. In other words, the chain is forced to relax its stress by reptation, that is curvilinear Brownian motion along the tube axis. Therefore, the unreleased stress in the system, G(t), is proportional to Ψ(t), the fraction of the initial tube that remains occupied by the chain at time *t*.

Originally, the tube model was designed to describe the motion of a long polymer chain in a fixed cross-linked rubber network [1]. Further development of the concept of reptation by Doi and Edwards [2] led to a full theoretical framework which is a cornerstone of the field of rheology of monodisperse linear entangled melts. To explain and fix some drawbacks in the predicted results several relaxation mechanisms were subsequently added to the reptation theory. For instance, the time it takes for the chain to diffuse out of the initial stressed tube $\tau_{d}$ was predicted to be proportional to the molecular mass *M* as $M^{3}$, while experimental data showed a $\tau_{d}∼M^{3.4}$ dependency [9]. The physical origin of this difference was explained later as some subsidiary relaxation process, which is faster than reptation [10]. This process is called primitive-path or contour-length fluctuations (CLF), according to which, due to Brownian motion, the ends of the chain can wrinkle quite some distance inside the tube, thereby vacating part of the tube end. When one end is pulled out back, it occupies a new random tube segment and part of the stress associated with the end of the original tube is lost. Thus, the contour length of the chain *L* fluctuates around the value <L>. Doi [11,12] performed the first calculation of CLF and analysed their effect upon reptation. The essential point of Doi’s arguments is that CLF reduces the reptation time $\tau_{d}$ for medium-sized chains (number of entanglements Z<100). His explanation is based on the idea that if the chain ends fluctuate rapidly over a distance ΔL, the chain needs to reptate only over a distance L−ΔL. As a result, the correction due to the contour length fluctuation is of the order of $1/\sqrt{Z}$ and the impact of the CLF reduces as the chain length increases. Different estimates of the CLF effect were later obtained by des Cloizeaux [13], Milner and McLeish [3], and Likhtman and McLeish [4]. An alternative view on the origin of 3.4 exponent for $\tau_{d}$ was also proposed by Liu et al. [14]. Their results indicate that the motions of surrounding chains have a significant influence on the latter dependency and the contribution of CLF to the $\tau_{d}$ scaling is quite weak.

Another mechanism, which needs to be taken into account, is the constraint release (CR) process, which emerges from the mobility of the surrounding chains. It is also included in the original tube theory [15]. Chains that form the tube of the “test” chain are moving as well in their own tubes. Thus, the tube itself evolves in time by releasing the topological constraints on the “test” chain and allowing the latter one to move on a large scale perpendicular to the tube axis. Marrucci [16] proposed that the process of constraint release speeds up the relaxation of the melt, considering that the relaxed portion of the chain behaves as solvent for the unrelaxed part of the molecule. CR is then modeled through “dynamic tube dilation” (DTD) [5,15,17] as well as “CR-Rouse motion” [15,18]. Within this framework, the relaxation modulus is given as $G\left(t\right)=G_{N}^{0}\Psi \left(t\right)\Phi^{\alpha }\left(t\right)$, where Ψ(t) is the fraction of the material which is not yet relaxed, and Φ(t) is the volume fraction which corresponds to the tube diameter such that $a\left(t\right)=a_{0}\Phi^{\alpha /2}\left(t\right)$. Under the assumption that an entanglement is formed by two chains, the dynamic dilution exponent α becomes 1. On the other hand, Colby and Rubinstein [19] suggest that α should equal 4/3 by considering that an entanglement is formed by some fixed number of binary contacts between the chains. It is important to note here that constraint release effects are also present in monodisperse samples. Indeed, as it was recently shown by Matsumiya et al. [20], viscoelastic relaxation process in the monodisperse system is accelerated by CR, and suppression of this mechanism is clearly observed if the chains are blended into a very long linear matrix. Thus, even for monodisperse samples, a good description of CR is needed in order to reach quantitative level of predictions.

All tube-based computational models that predict the linear viscoelasticity (LVE) agree that, in addition to the high frequency Rouse modes, the relaxation of linear polymer melts consists of the three main processes described above: reptation, contour length fluctuations and constraint release. For example, these mechanisms form the basis of the quantitative theory developed by Milner and McLeish (MM theory) for monodisperse linear polymers [3]. In this model, a linear chain is represented as a two-arm star polymer able to reptate. In this way, relaxation starts with contour length fluctuations and retraction (activated CLF) of the chain ends, and at time $t=\tau_{d}$ is followed by reptation along the unrelaxed part, as retraction is becoming slower than reptation. The Hierarchical model of Larson [6] incorporates the physics of the MM theory and together with some additional features forms a model which can be applied to linear polymers as well as to mixtures of branched polymer molecules. This model was modified by Park [21,22,23] to include the “early-time” fluctuations and some other refinements. The BoB model, which also uses the MM theory, was developed by Das et al. [7] and incorporates modeling of branch on branch structures.

While all models use a common theoretical approach for the LVE of linear entangled polymers, they differ in some important assumptions [24]. Notably, the definition of CLF in models which are based on the MM theory, contradicts the original idea developed by Doi [2], according to which only the fast fluctuations at the end of the chain shorten the contour length of the tube. Therefore, although theoretically it is based on Doi’s assumption [12], numerically the amount of the material that is being subtracted from the reptation time seems to be overestimated. The effect of this assumption is discussed in detail in Section 4. On the other hand, as is discussed in Section 5, the MM theory neglects the constraint release modes arising from reptation. In the present work, we investigate the importance of this modeling choice and its impact on the predictions of LVE by comparing the results obtained with the BoB model to a large set of experimental data. Tube models also require the definition of an expression for the relaxation modulus. The BoB and the Hierarchical models use the same expression as the one used in the MM theory. In Section 6, we discuss the accuracy of this expression, and propose an alternative derivation of this relaxation modulus.

Another LVE theory was proposed by Likhtman and McLeish [4] (LM theory). It involves all the relevant mechanisms considered above and does not include any new physical processes. This theory solves the existing mathematical models for the relaxation processes in a consistent and more accurate way. In particular, the expression for the amount of unrelaxed material at early times is derived by solving the one-chain problem combining analytical and stochastic approaches. The resulting function slightly enforces the influence of early fluctuations, as compared to the one used in the other models, and the transition between CLF and reptation regimes always happens at time $3.6\tau_{R}$, where $\tau_{R}$ is the Rouse time. As discussed in Section 5, this estimate is much closer to the original idea of CLF, than the estimates employed in the BoB and Hierarchical models.

As a third approach, the time-marching algorithm (TMA) [8] for monodisperse linear polymers is also a tube-based model, but in fact, it is quite different from the other models. For example, CLF and reptation processes act simultaneously, while the MM, Hierarchical and BoB models consider reptation and CLF as mutually exclusive processes with a transition time at $t=\tau_{d}$. From cursory analysis, there seems to be a substantial distinction between TMA and other models. For instance, reptation is often a slower process in TMA, however arm fluctuation can speed up the terminal relaxation indirectly through Ψ(t), which is the fraction of material not yet relaxed by reptation or retraction. It must be noted that, in addition to these relaxation processes, all above models consider the high frequency Rouse relaxation in a similar way [4].

Therefore, a first aim of this paper is to discuss the main differences between TMA approach and the models based on the MM theory, as well as the influence that each assumption has on the relaxation process. In particular, we would like to understand why all these models, despite their large differences, lead to similar predictions in case of linear polymers. To this end, we collected a large set of experimental data from literature, that we use, first, to discuss the material parameters of the different polymers, then to validate the different approaches. This comparison allows us to address two important questions: (1) what is the importance of CLF in the relaxation process of a linear chain? (2) how does it affect the terminal relaxation time of the chains? Based on our conclusions, we then propose to modify the TMA model. Another purpose of this work is to draw attention to the differences observed between the theory and the numerical implementation for the BoB and the Hierarchical models.

From a more general perspective, the objective of the present work is to revisit the existing tube models and propose improvements in order to be able to extend them towards a quantitative prediction of the viscoelastic properties of polydisperse polymers. Indeed, while this work only focuses on monodisperse linear polymers, it allows us to discuss in depth three issues which are intrinsic to the models, and which can have very large consequences on the prediction of the relaxation of more complex polymers.

The first question is related to the balance between reptation and fluctuation processes, which is of primary importance if we want to be able to predict the viscoelastic behavior of both linear and star-like polymers while keeping the same parameters [25]. And this first step is of primary importance if we want to be able to predict the relaxation behavior of binary blends composed of linear and star-like polymers. As mentioned in ref. [26], today, most of the tube models fail at doing so, which is probably the strongest limitation for applying such models to polydisperse samples.

The second question that we would like to address here is the too fast relaxation of very short polymer chains [27], which leads to a wrong description of the plateau modulus [14,28,29]. This requires a careful description of the chain dynamics at short times, in order to prevent the possible disentanglement of the chains before the Rouse relaxation of their entanglement segments.

The third question is related to the influence of constraint release (CR) process, and how this one is taken into account in the models. As already mentioned, its effect is quite large, even in the case of monodisperse linear polymers. By combining the present results obtained for monodisperse linear polymers with the ones coming from recent studies on the viscoelastic properties of binary blends of entangled linear polymers [30,31,32,33,34], in which mechanisms such as the Constraint Release Rouse process and the dynamic tube dilution have been studied in detail, we believe all the ingredients will be there in order to extend these studies to the linear viscoelastic properties of polydisperse samples.

This paper is organized as follows. First, in Section 2, we present the time-marching algorithm (TMA) for the case of linear macromolecules. Second, in Section 3, we present the experimental data together with the material parameters used within the present study. In Section 4 and Section 5, we compare the theoretical predictions of TMA with LVE data of polybutadiene, polyisoprene and polystyrene polymer samples and propose a novel modification to the model. Then, in Section 6, we discuss the main differences between the TMA and the BoB model, focusing on the way CLF is treated, and on the way the relaxation modulus is derived. Additionally, we propose changes to the BoB model, so that the implementation is consistent with the theory. Section 7 compares the results of the TMA and BoB predictions for the various sets of experimental data, and finally conclusions are presented at the end of the article.

## 2. Modeling

The TMA model for prediction of the LVE properties of linear polymers is described in several publications [8,34], however in slightly different forms. In the current paper, we present a new version of the TMA, which is consistent in both definitions and notation with our earlier work [25] on star-shaped polymer melts, as well as with tube theory for linear polymers.

### 2.1. Modeling of the Relaxation Modulus in TMA

As already mentioned, all tube-based models assume that a single polymer is represented by an ideal chain (i.e., a random walk), which is trapped in a so-called tube. Such a polymeric system will always remain in equilibrium, unless it is forced to change its state. If the system is subjected to a strain, the distribution of chains will be disturbed, and the polymer melt is under stress. The system should readjust its “stressed” configuration in order to reduce that stress. Each chain has to move out of its original “stressed” tube into a new tube, through a relaxation mechanism such as reptation, CLF and retraction. Every time a portion of the tube is vacated by the chain, the memory of the orientation is lost. This memory loss is described by the survival function Ψ(t), defined as the unrelaxed fraction of initial tube segments at time *t*.

There are several time-scales involved in the relaxation process. At a very short time scale, less than the Rouse time of an entanglement segment $\tau_{e}$, the chains are unaware that they are surrounded by other chains and thus, they relax locally through Rouse motion. The chain starts to feel the topological constraints due to its surroundings only after $\tau_{e}$. This prevents the polymer chain from moving perpendicular to the tube axis. At this time, since we treat the linear chain as a two-arm star, the stress starts to decrease through CLF and arm retraction. Nevertheless, it is not the only possible relaxation processes taking place at time $t>\tau_{e}$. Indeed, unlike other models, the TMA assumes that the reptation and retraction processes are independent and can occur simultaneously. Recall that, the “test” chain is a statistical representation of all polymers in the system. Furthermore, as discussed elsewhere [25], the effect of constraint release must be taken into account via the concept of “dynamic tube dilation” [3,15,16]. According to this picture, the relaxed portion of the arm acts as a solvent for the unrelaxed part, thereby reducing the entanglement density, i.e., increasing the effective tube diameter, and finally speeding up stress relaxation. Nonetheless, since in monodisperse linear melt all chains relax nearly simultaneously these last ones are assumed to reptate in an undilated tube. All three relaxation mechanisms and their relationships are discussed in detail below.

The relaxation of a polymer after a small step strain is usually described by the relaxation modulus G(t), which is the ratio of the stress remaining at time *t* to the applied step strain [2]. The stress relaxation modulus of any polymer melt includes contributions of each relaxation mechanism:(1)$$\begin{array}{cc} G(t) & =G_{N}^{0}F\left(t\right)+F_{LR}+F_{FR}, \end{array}$$(2)$$\begin{array}{cc} F(t) & =\Psi \left(t\right)\Phi {\left(t\right)}^{\alpha }, \end{array}$$
where Ψ(t) is the fraction of material not yet relaxed by reptation or retraction, Φ(t) is responsible for CR and defines the volume fraction of the chains acting as effective constraints in the system, α is the dynamic tube dilution exponent, $F_{LR}$ is the contribution of the longitudinal Rouse modes [4] and $F_{FR}$ is that of unconstrained fast Rouse motion.

After the Rouse relaxation of the entanglement segments, the macromolecules are coarse grained at the level of segments between neighbouring entanglements, and they are represented by their primitive path. The location of a primitive chain segment is represented by the normalized curvilinear variable *x*, ranging from 0 at the chain end to 1 at the center of the chain. The segment survival probabilities are determined for each time step $t_{i}$. Let $p_{fluct}\left(x,t_{i}\right)$ and $p_{rept}\left(x,t_{i}\right)$ be the probabilities that the segment *x* of the primitive chain remains inside the initial tube at time $t_{i}$ and is not yet relaxed by fluctuation or by reptation, respectively. As a result, the relaxation function Ψ(t) is calculated by summing up the survival probabilities of all segments along the chain:(3)$$\begin{array}{cc} \Psi (t) & =\int_{0}^{1}p_{fluct}\left(x,t\right)p_{rept}\left(x,t\right)dx. \end{array}$$
Thus, the probability of the segment *x* to survive relaxation at time *t* is the joint probability of two separate events: surviving from the relaxation either by retraction or by reptation. In contrast to the other models, we assume that the prevailing relaxation process within the polymer system is defined implicitly via the probability values. For example, we can observe a clear separation between retraction and reptation only for a long chain. Basically, each chain in a melt has a possibility to relax its segment *x* via one of the relaxation mechanisms to which a probability is assigned.

On the other hand, the dilution factor Φ(t) is usually equal to the fraction of the survived initial entanglements Ψ(t). However, if a large fraction of the material is relaxed in a very short time window, the tube widening becomes faster than the rate at which chains are able to explore it at that time. Therefore, in such case, the “constraint-release Rouse motion”(CRR) [3] process controls the limits of the tube diameter, in the following way:(4)$$\begin{array}{c} \Phi_{CRR}\left(t_{i}\right)=\Phi \left(t_{i-1}\right){\left[\frac{t_{i-1}}{t_{i}}\right]}^{1/2\alpha }, \end{array}$$
(5)$$\begin{array}{c} \Phi \left(t_{i}\right)=max\left\{\Psi \left(t_{i}\right),\Phi_{CRR}\left(t_{i}\right)\right\}, \end{array}$$
where $\Psi (t_{i})$ is the the unrelaxed volume fraction at time $t_{i}$ and $\Psi_{CRR}\left(t_{i}\right)$ defines the diameter of the supertube that could be explored by the Rouse motion of the confining tube at time $t_{i}$. Out of the CRR regime, $\Phi \left(t_{i}\right)=\Psi \left(t_{i}\right)$.

The last terms of the relaxation modulus are the contribution from the high-frequency Rouse relaxation modes. They include stress relaxation of segments of the test chain both shorter than $\tau_{e}$, and longer than $\tau_{e}$, during which only the longitudinal Rouse modes along the tube [4]:(6)$$\begin{array}{cc} F_{FR} & =G_{N}^{0}\frac{5}{4Z}\sum_{j=Z}^{+\infty }\exp\left(\frac{-2j^{2}t}{\tau_{R}}\right), \end{array}$$
(7)$$\begin{array}{cc} F_{LR} & =G_{N}^{0}\frac{1}{4Z}\sum_{j=1}^{Z-1}\exp\left(\frac{-j^{2}t}{\tau_{R}}\right), \end{array}$$
are active. In the above, $Z=M/M_{e}$ is the number of entanglements per chain and $\tau_{R}=Z^{2}\tau_{e}$ is the Rouse time.

### 2.2. Contour Length Fluctuations in TMA

As it is mentioned above, TMA considers reptation and fluctuations as independent processes, which are never interrupted during the polymer relaxation and can occur at the same time. Furthermore, in the TMA model, the CLF do not have an explicit effect on the disentanglement time $\tau_{d}$: there is no shortening of the distance along which the chain must reptate. Nevertheless, for the chains which are not too long (Z<30), there is an implicit impact of the arm retraction on reptation via the product of the survival probabilities (see Equation (3)). It must be noted that in such a case, the segments close to the middle of the chain (0.5<x<1) have approximately equal opportunity to be relaxed by retraction or by reptation ($p_{fluct}\left(x,t\right)≃p_{rept}\left(x,t\right)<1$).

In order to visualize the qualitative behaviour of TMA, one should consider that at time *t*, the test chain represents the average behaviour of the whole system. Thus, the relaxation state of a chain segment *x* reflects the statistical value of the segment relaxation, which is determined from the probability distribution over all possible relaxation states of this molecular segment of all chains in the system. At the beginning, most of the chains renew their outer segments via retraction as this is much faster than reptation. Later, the diffusion (reptation) of the middle segments catches up with the speed of retraction, such that $p_{fluct}\left(x,t\right)\geq p_{rept}\left(x,t\right)$ for unrelaxed segments *x*. Thus, the survival probabilities of these segments are $p_{fluct}\left(x,t\right)≃1$ for long chains, and $p_{fluct}\left(x,t\right)≃p_{rept}\left(x,t\right)$ for shorter ones. Therefore, most of the chains with high molecular weight end their relaxation only with reptation, while relatively smaller chains have a possibility to release their final segments from the initial tube via both reptation and retraction.

The survival probability $p_{fluct}\left(x,t\right)$ of a given segment *x* decays with time and is calculated by the formula:(8)$$\begin{array}{c} p_{fluct}\left(x,t\right)=\exp\left(\frac{-t}{\tau_{fluct}\left(x,t\right)}\right), \end{array}$$
where the characteristic fluctuation time $\tau_{fluct}\left(x,t\right)$ is defined by:(9)$$\begin{array}{cc} \tau_{fluct}\left(x,t\right) & =\tau_{early}\left(x\right),\;x\leq x_{tr}, \end{array}$$
(10)$$\begin{array}{cc} \tau_{fluct}\left(x,t\right) & =\tau_{early}\left(x_{tr}\right)\exp\left(\frac{U\left(x,t\right)-U\left(x_{tr},t\right)}{k_{B}T}\right),\;x>x_{tr}. \end{array}$$
The survival probability needs to be recalculated at every time step for all unrelaxed segments *x*, whereas the reptation time $\tau_{rept}$ remains constant.

In TMA, the transition between the two fluctuation processes happens at the transition segment $x_{tr}$, for which the potential is of the same order as the thermal energy $\left(U\left(x_{tr}\right)\approx k_{B}T\right)$. The potential is irrelevant for the early fluctuations; the end of the chain can move freely, and the relaxation time is estimated by:(11)$$\begin{array}{c} \tau_{early}\left(x\right)=\frac{9\pi^{3}}{16}\tau_{e}{\left(\frac{Z}{2}\right)}^{4}x^{4}. \end{array}$$

Deep retractions are entropically unfavourable and are represented by the diffusion of the chain end uphill in an effective potential
(12)$$\begin{array}{cc} U(x,t) & =\frac{3}{2}k_{B}TZ\left(\int_{0}^{x_{c}\left(t\right)}\xi \Phi {\left(\tau \left(\xi \right)\right)}^{\alpha }d\xi +\Phi {\left(t\right)}^{\alpha }\int_{x_{c}\left(t\right)}^{x}\xi d\xi \right),\;x>x_{c}\left(t\right). \end{array}$$

In the above equation, Φ(τ(ξ)) is the polymer fraction not relaxed at time τ(ξ) when segment ξ was relaxed and $x_{c}\left(t\right)$ is the segment relaxing at the current time *t*, such that $t=\tau (x_{c})$ [25].

As a remark, it should be noted that in our recent paper [34] the modeling of the arm retraction excludes the “deep” fluctuations, as we tried to follow the common understanding that reptation should be always faster than an exponentially suppressed process and CLF includes only fast retraction. Definitely, at some point reptation takes over from the slowing down fluctuations. However, there is no agreement yet on the estimation of the transition zone between the two relaxation processes. Until now, all theories, except LM, include slow retraction in their modeling of CLF. Moreover, MM-based theories assume that deep fluctuations in the contour length of the linear polymer along its tube will shorten the distance of reptation. Arguably, this seems to be a too strong modeling assumption as discussed in Section 5.

### 2.3. Reptation

The probability that the segment *x*, going from one chain end (y=0) to the other extremity (y=1), is not yet relaxed by reptation at time *t* was determined by Doi-Edwards [2] to be:(13)$$\begin{array}{c} p_{rept}\left(y,t\right)=\sum_{podd}\frac{4}{p\pi }\sin\left(\frac{p\pi y}{2}\right)\exp\left(\frac{-p^{2}t}{\tau_{d}}\right), \end{array}$$
where the reptation time is given by:(14)$$\begin{array}{c} \tau_{d}=3\tau_{e}Z^{3}{\left\{\Phi^{*}\right\}}^{\alpha }. \end{array}$$

In the coordinate system used by TMA, we only calculate the relaxation times of the segments for half of the chain (from x=0 at the chain end, to 1 in the middle), so that y=1−x/2. The volume fraction $\Phi^{*}$ defines the diameter of the tube in which the chain reptates. In this work, since only monodisperse polymers are studied, we assume that the polymer chain reptates in the undilated tube. This is equivalent to $\Phi^{*}=1$.

## 3. Experimental Section

In the subsequent discussion, we present and analyse the experimental linear viscoelastic response of thirteen sets of monodisperse linear polymer samples.

### 3.1. Materials

The molecular characteristics of experimental samples used in this work are listed in Table 1, Table 2 and Table 3. The number of entanglements *Z* has been obtained by using the values of $M_{e}$ from Table 4. 


*Polybutadiene*


Different set of polybutadiene samples were analyzed. The first two sets of data were collected from Kapnistos et al. [35] and Liu et al. [14], except for PBD165, which was synthesized by Roovers [36]. The data of Baumgaertel et al. [37] and Lee et al. [38] has been used in the paper of Likhtman and McLeish [4] to validate the theoretical predictions based on their model. The authors concluded that different values of $M_{e}$ and $\tau_{e}$ must be used for each set of data. At this point, it seems important to underline that in the work of Likhtman and McLeish [4], data referencing is questionable. Indeed, the authors refer tot the data from the work of Juliani [39], while in reality, they used unpublished data from the group of Archer [40]. Part of that data was reported later by Lee et al. [38]. In this work we compare all above mentioned literature data to examine the consistency between the input parameter values ($M_{e}$, $\tau_{e}$) used for TMA and BoB predictions.


*Polyisoprene*


The viscoelastic measurements for polyisoprene (PI) samples were taken from Auhl et al. [41] and Watanabe et al. [42]. The data is listed in Table 2. It must be noted here that the reference temperatures for both data sets are different.


*Polystyrene*


The sets of polystyrene (PS) viscoelastic data are coming from Huang et al. [43], Nielsen et al. [44] and Shahid et al. [45]. In addition, we use experimental data from Schausberger et al. [46] and Graessley et al. [47], which were also analysed in the work of Likhtman and McLeish [4]. The data can be found in Table 3. To fit the data accurately, they had to adjust the values of $M_{e}$, $\tau_{e}$ and $G_{N}^{0}$ separately for each set. As it appears later, this is not the case for TMA and BoB.

### 3.2. Material Parameters

In order to test TMA and BoB against the experimental data, we need, first, to define several tube parameters: the plateau modulus $G_{N}^{0}$, the molecular weight between two entanglements $M_{e}$, the Rouse time of an entanglement segment $\tau_{e}$ and the dilution coefficient α. It is considered [48], that the plateau modulus and entanglement molecular weight are related to each other by $G_{N}^{0}=\left(4/5\right)\left(\rho RT/M_{e}\right)$, with ρ being the density of polymer, *R* the universal gas constant, and *T* the absolute temperature. While $M_{e}$ and $\tau_{e}$ are material and temperature dependent, the dilution exponent α is expected to be independent and has been set to 1 for all predictions.

As a first point, we show that any molecular theory used for describing the linear viscoelasticity of monodisperse polymers should be independent of the chemistry of the samples. Under such assumption, we normalized the storage and loss moduli of several sets of data by a horizontal shift factor $\tau_{e}$, and by a vertical shift factor equal to $1/G_{N}^{0}$ (see Table 4). In Figure 1, we present the normalized data of specific PBD, PS and PI polymers, which show good superposition at high and low frequencies. With respect to the tube model, this means that these polymers have an identical number of entanglements per molecule, *Z*, since they exhibit similar LVE behavior in the sense that the shape of the $G^{\prime }$ and $G^{\prime \prime }$ curves plotted against the angular frequency ω is the same.

For convenience, we report the length of the chosen samples, as well as the number of entanglement segments, using the $M_{e}$ values from Table 4.

Therefore, from the similarity of the curves, we can conclude that the pairs of polymers in Figure 1 have identical ratio $M/M_{e}$. Since their molar masses are known, this allows us to take one chemical type of samples as a reference, for example PS, and determine the relative values of $M_{e}$ for other samples (PBD, PI) by looking at the ratio between their molar masses.

We fix $M_{e}$ = 15,350 g/mol for PS samples for different reference temperatures, since the same value was used in our previous work on star-shaped polymers [25]. Now, looking at the comparison of PS390 versus PBD47, it seems that the ratio $M_{e}^{PS}/M_{e}^{PBD}$ must be approximately 7.6. Thus, based on the value taken from PS, the value of $M_{e}$ for PBD is expected to be close to 2000 g/mol. In case of Auhl’s PI samples, by comparing (PS100-PI33) and (PI90-PBD39), we find that the ratios are $M_{e}^{PS}/M_{e}^{PI}=3.1$ and $M_{e}^{PI}/M_{e}^{PBD}=2.5$. Therefore, $M_{e}^{PI}$ should be around 4900 g/mol, while $M_{e}^{PBD}$ again shows a value in the vicinity of 2000 g/mol. However, comparing the data of Watanabe (PI43-PBD21) we get a different result, $M_{e}^{PI}/M_{e}^{PBD}$ slightly smaller than 1.9, which leads to $M_{e}^{PI}$ slightly shorter than 3800 g/mol. While the origin of this difference is unclear, this last value of $M_{e}$ is consistent with the one used in our previous paper [25]. Based on this observation, two different sets of material parameters are used for modeling the LVE of the PI samples. The exact values of $M_{e}$ are found in Table 4. They were slightly adjusted, based on the best fitting of storage and loss modulus for all samples.

In Figure 1 we also compare rheological data of polybutadiene chains coming from two different sources: Baumgaertel et al. [37] (PBD97) and Liu et al. [14] (PBD98). Recently, this data has been analyzed by Park et al. [49]. The authors concluded that the data of Baumgaertel et al. [37] do not follow the same relaxation behaviour as the other linear PBD. However, as it can be seen from Figure 1, the viscoelastic data of PBD98 and PBD97 coincide perfectly at the transition frequency.

Besides $M_{e}$, the value of $\tau_{e}$ must be determined. The choice of $\tau_{e}$ is driven by the idea to have the best fit between the data at high-frequencies, where the Rouse process takes place. The same value of entanglement time should be used in both TMA and BoB model, as the two approaches use the same description of longitudinal relaxation and fast Rouse relaxation [7]. In general, $\tau_{e}$ should have a universal value for samples at the same temperature and can be extracted from the high-frequency data irrespective of the molecular weight or architecture. This holds only when determination of the shift factors for the time-temperature superposition is performed consistently and the glassy dynamics does not affect the experimental measurements at those frequencies (not included in the modeling). Based on latter statement, Park et al. [49] found that the equilibration time for PBD should be $\tau_{e}=\left(3.7\pm 0.93\right)\times 10^{-7}$ s at 25 °C. In the present work all predictions for PBD samples were performed by considering the Rouse time of an entanglement segment as $\tau_{e}=4.0\times 10^{-7}$ at $T\;=\;26\pm 2{\;}^{\circ }C$. This is thus in good agreement with the work of Park et al. [49].

The third parameter to define is the plateau modulus. It should follow the definition of Fetters [50] and be given by $G_{N}^{0}=\left(4/5\right)\left(\rho RT/M_{e}\right)$. In order to determine it, we use the following densities of polymer melts: ρ=0.91 g/cm^3^ (PI 25 °C ), ρ=0.90 g/cm^3^ (PI 40 °C), ρ=0.9 g/cm^3^ (PBD) [51], ρ=1.01 g/cm^3^ (PS 130 °C) and ρ=0.98 g/cm^3^ (PS 180 °C). Note that the temperature dependencies of the density for PI and PS that we use are given by $\rho \left(T\right)=0.9283-0.61\times 10^{-3}T$ and $\rho \left(T\right)=1.2503-6.05\times 10^{-4}\left(273.15+T\right)$, respectively [52,53]. The calculated values of $G_{N}^{0}$ can be found in Table 4. They are generally quite low compared to the values usually used. If we compare them to the values found by the best fitting, they are by ∼20% lower (except for Watanabe’s and Baumgaertel data), i.e., closer to the definition proposed by Ferry [9] for $M_{e}$, where the prefactor of 4/5 is omitted (see Table 4). In order to use the Fetters definition, one would need to set α to 4/3, which would allow us to take smaller values of $M_{e}$[21,22,24,38]. Alternatively, reducing the equilibration time $\tau_{e}$ by a factor 2 would also lead to the choice of a smaller $M_{e}$ values. However, such choice requires corrections in high-frequency predictions of Rouse process. This idea is discussed in a recent paper of Park et al. [49]. Here, we accept the arguments of Auhl et al. [41], that many mechanisms can affect the prefactor in the definition of $M_{e}$. Thus, the precise value of $G_{N}^{0}$ is unknown, and we use the one obtained from the best fitting. Data from Auhl et al. has been already modeled by LM theory [41], using the following input parameters for PI: $M_{e}=4820$, $G_{N}^{0}=0.47$ and $\tau_{e}=1.32\times 10^{-5}$. Those values are comparable with the ones used in the TMA model. Moreover, experimental data available for PBD requires the parameters similar to the ones determined by Roovers for PBD165 sample [36] ($G_{N}^{0}=1.07$, $\tau_{e}=2.56\times 10^{-7},M_{e}=1980$). PBD data from Baumgaertel requires a higher value of $G_{N}^{0}$ as usually accepted, however it is consistent with the values used for the same samples by Baumgaertel et al. [37] and Likhtman et al. [4].

The resulting model parameters are listed in Table 4.

## 4. Renormalization of the Coordinate System

In our previous study [25], we propose a renormalization of the segment coordinates in order to improve LVE predictions of the TMA model for short star-shaped polymers (arm length Z<20). Indeed, we noticed that by time $\tau_{e}$, a large portion ($0.48M_{e}$) of the shorter chains has already been relaxed due to CLF. A similar conclusion was made by van Ruymbeke et al. [35] while investigating discrepancies in tube-model predictions regarding the plateau modulus of short linear chains. This fast relaxation has also been reported in several works, such as the work by Liu et al. [14,28] and by Hou et al. [29] when comparing the LM theory to molecular dynamics (MD) simulations of bead-spring polymer melts. The early relaxation, observed at time smaller than $\tau_{e}$, predicted by the tube-based models contradicts the tube picture, since, at time $t<\tau_{e}$ , chains should only relax by Rouse motion, and are unaware of the surrounding chains. However, classical tube-based models consider that relaxation by CLF starts immediately at t=0. This effect is especially important for short chains, as observed by Bacova et al. [54] in their MD simulation data analyses of the entangled linear and star polymers. Furthermore, the latter study reveals that the primitive path near the chain ends is almost fully relaxed by simple Rouse dynamics at $t<\tau_{e}$, since the outer segments are weakly affected by the topological constraints. Therefore, the decay of CLF at time earlier than $\tau_{e}$ should be omitted, since it is already taken into account by the high-frequency Rouse relaxation. Otherwise, the chain relaxation is underestimated and the level of the plateau modulus is predicted too low. This assumption strongly affects the relaxation of short molecules, and especially star-shaped molecules since segmental fluctuation times are exponentially dependent on the square of their depth and the unrelaxed fraction [25].

In fact, many works on poorly entangled chains have shown that the dynamics of the outer segments should be treated differently than the dynamics of the middle part of a chain due to the variant nature of entanglements. It seems that the chain tips show a Rouse-like relaxation. Here, we briefly review these works, before discussing how tube models can be corrected in order to avoid double counting of the relaxation of the outer segments (i.e., by Rouse and CLF). This issue was well described by Kavassalis and Noolandi [55,56,57], who presented a packing entanglement theory, where the loose chain ends are not expected to contribute to the formation of effective entanglements. High mobility of the chain ends prevents them from forming long-living connections with other chains. According to their model, an effective entanglement is defined by the presence of a certain number of non-tail neighboring polymer segments; the concentration of the latter segments scales as λ=1−1/Z. The average tail length is $M_{t}\approx 0.5M_{e}$. Thus, as chains get shorter, more and more dangling ends are excluded from contributing to entanglements, and the entanglement spacing $M_{e}$ varies with molecular weight *M*. To support this assumption, Rathgeber et al. [58] used neutron spin-echo spectroscopy (NSE) to investigate on a microscopic length scale the influence of topological constraints on the single-chain dynamics. To this end, they varied the mass fraction of long tracer chains in a short-chain matrix. As a result, the authors found that the number of end segments relative to the number of inner segments increases with decreasing tracer fraction, thus weakening the topological constraints. Consequently, the value of $M_{e}$ and the concentration of longer chains are directly proportional, since shorter chains create fewer entanglements. Taking chain-end effects into account, Rathgeber [58] obtained an excellent description of the NSE data.

In addition to rheology and NSE measurements, nuclear magnetic resonance (NMR) T${}_{2}$ relaxation time studies [59,60] can provide information on the chain mobility. Most NMR experiments measure segmental dynamics ultimately in terms of an orientation autocorrelation function [61]. Highly mobile polymer chain ends exhibit an isotropic motion (fast reorientation), which shows a slow transverse relaxation with a long corresponding relaxation time T${}_{2}$. At the same time, the entanglement network induces an anisotropy of chain motion on the central segments. Segments located near chain ends are expected to be subject to molecular motions different than those in the central block, provided that the molecular mass of the chain is above the critical value $M_{c}$, in order to ensure entangled dynamics. As far as NMR is concerned, Kimmich et al. [62] proposed that the outer entanglement lengths of a reptating chain behave in a similar fashion to a Rouse chain not subjected to topological constraints. Indeed, through analysis of bi-exponential fits to experimental data, they showed that the effective molecular weight of the end performing Rouse motion is independent of the molecular weight of the chain, and approximately equal to $M_{e}$. This is consistent with the fact that unentangled linear chains with a molecular weight approaching $M_{c}$, display Rouse dynamics.

Moreover, also based on NMR, Guillermo and Addad [63] showed that it is possible to represent a chain as two end-submolecules, independent of *M*, and one inner part with clearly distinct dynamics. The idea that the end sections are undergoing fast Rouse-like dynamics, whereas the central section is undergoing slower dynamics is also confirmed by Ries et al. [64] and by Chavez and Saalwachter [61]. The latter research showed that during the constraint Rouse regime the chain is less restricted than predicted by the tube model. Wang et al. [65] argue that this conclusion reflects the importance of ruling out the chain end contributions. Nevertheless, today, the main problem is that the transition between isotropic and anisotropic parts cannot be clearly defined [60,66]. Furthermore, it is not completely clear to which extent the constrained Rouse motion is observed by NMR as isotropic or not. However, it is clear from the above that it is important to separate contribution of end monomers to the relaxation mechanism and include it to the current TMA model, even if quantitative description of chain-end effect still needs to be refined by NMR experiments or computer simulations.

A first solution was proposed by van Ruymbeke et al. [35], in order to slow down the CLF process: (1) adding to the chain an abstract sub-chain of length $0.48M_{e}$ , thus forming an “equivalent” chain, and, (2) rescaling the coordinate system. As a result, the fluctuation time of the original chain end is equal to $\tau_{e}$ in the new coordinate system of the “equivalent” molecule. The obtained result is in good agreement with the experimental data for short linear chains. Nevertheless, this procedure is rather a numerical technique for being consistent with the tube theory. Instead, to attain consistency with our previous works, we propose here to account for the role of chain ends in the relaxation process through three considerations: (1) effective entanglements are only formed by the chain segments with relaxation time higher than $\tau_{e}$, in other words, chain ends are not part of the entanglement network due to their high mobility, (2) they relax via free Rouse-like motion, and therefore, (3) they are neglected in the calculations of the relaxation of the chain by fluctuations or reptation. To this end, we introduced [25] a new coordinate system with the origin x=0 localized at the molecular segment $x_{0}$, which has a relaxation time $\tau_{e}$. This correction is in qualitative agreement with the approach of van Ruymbeke et al. [35], while it includes the specific influence of chain ends on the entanglement network as the chain length is shortened. In this way, the latter effect makes our approach consistent with the tube theory.

The above mentioned discrepancies in tube-model predictions regarding the plateau modulus of short linear chains are illustrated in Figure 2. In this figure, the original TMA predictions (dash-dotted line) of the storage modulus are compared to experimental results for short PBD polymers. The results are obtained with material parameters from Table 4. Whereas the agreement between theory and experiments is relatively good, the plateau modulus is poorly described for really short chains, like PBD8 and PBD21, due to the fact that the chain starts to relax their tube before $\tau_{e}$. Hence, using the new coordinate system in order to prevent this too fast relaxation seems important.

In our work on star-shaped polymers [25], we showed that part of the tube end of length $s_{0}=0.48M_{e}$ is relaxed by time $\tau_{e}$ via the free Rouse motion. This means that the relaxation of the ends is already included in high-frequency Rouse relaxation defined by Equation (6) and must be excluded from the determination of the relaxation function Ψ(t) in Equation (3). Here, we use the same result, which is in good agreement with the estimation provided by the theory of Kavassalis and Noolandi [55,56,57]. This requires working in a new coordinate system, as already mentioned.

Indeed, once relaxation reaches $\tau_{e}$, the original tube with equilibrium length $L_{0}$ will already lose its ends up to segment $x_{0}$, and the molecular weight between entanglements needs to be renormalized as $M_{e}^{*}=M_{e}/\left(1-x_{0}\right)$. The segmental Rouse time is also rescaled as $\tau_{e}^{*}=\tau_{e}/{\left(1-x_{0}\right)}^{2}$ and the Rouse relaxation process is postponed till $\tau_{e}^{*}$, such that segments up to $x_{0}^{*}$ are relaxed. Thus, $x_{0}^{*}$ represents the deepest segment which relaxes by high-frequency Rouse relaxation. Segments after $x_{0}^{*}$ are relaxing in a tube. Therefore, after a time $\tau_{e}^{*}$, only the central part of the chain with molecular weight of $M(1-x_{0}^{*})$ is confined in the tube. This means that the relaxation function Ψ(t) is calculated over the interval $\left[x_{0}^{*},1\right]$. Based on the above arguments, following on previous work [25], retraction and reptation should be calculated in a new coordinate system with the origin $x_{eq}=0$, localized at the tube segment $x_{0}^{*}$, which has a relaxation time $\tau_{e}^{*}$ in the original coordinate system. However, to keep the calculations simple and due to negligible difference in predictions made using $x_{0}^{*}$ and $x_{0}$, we start the new coordinate system at $x_{0}$ as indicated in Figure 3. The same holds for $\tau_{e}^{*}$, where $\tau_{e}$ is used instead.

Following the classical tube theory, early fluctuations are not influenced by dilution. The relaxation time $\tau_{early}$ is found for x∈[0,1], but only values from $x_{0}$ are valid for the calculations. Thus, for the new coordinate system Equation (11) should be rescaled as
(15)$$\begin{array}{c} x=x_{0}+x_{eq}\left(1-x_{0}\right),\;x_{eq}\in \left[0,1\right]. \end{array}$$

The above renormalization ensures that at time $t=\tau_{e}$ the tube in the new coordinate system is yet to start relaxing. Additionally, it only affects the properties of the short chains (Z<20). Results obtained with the new coordinate system are shown in Figure 2 (dashed line). The agreement between the experimental plateau modulus of short linear chains and the theoretical one is certainly improved. It must be noted that in our initial work on linear binary blends [34] the prefactor 0.5 in the fluctuations process of TMA could have been avoided by considering the coordinate transformation under discussion, along with considering the effects of deep retraction process.

## 5. Comprehension of CLF in the Tube Models

Besides the importance of avoiding CLF before $t=\tau_{e}$, another important question to address is the transition between fluctuations and reptation processes, and how it is taken into account in the different models. To this end, we first discuss the comparison between the experimental and theoretical PBD viscoelastic data presented in Figure 2. For the theoretical predictions of $G^{\prime }$ and $G^{\prime \prime }$ calculated by the TMA model with the coordinate renormalization as described in Section 4, we see that the modeling results (dashed line) correctly predict the plateau modulus, but show a too slow relaxation in the terminal zone for the shorter polymers.

This deviation can be attributed to the fact that the influence of CLF process on reptation is neglected. Indeed, although the TMA model incorporates the retraction of the chain ends, this effect is not accounted for as tube length fluctuations that speed up reptation. This contradicts, the original idea behind CLF [2,11,12], which states that only in the absence of fluctuations the disengagement time is estimated as the time $\tau_{d}=\frac{L^{2}}{D}$ necessary for the chain to move by distance *L*. In case of fluctuations, the chain can disengage from the tube when it moves the distance $L_{eff}=L-\Delta L$. However, only under the condition that the chain ends are fluctuating rapidly over the distance ΔL. Consequently, the disengagement time should be $\tau_{d}=\frac{L_{eff}^{2}}{D}$[2]. While, in TMA, the reptation time in Equation (14) involves the entire chain length *L*, most models consider a linear chain being relaxed by reptation, when its center of mass diffuses along the tube over a shorter distance $L_{eff}=L-\Delta L$. In the case of MM-based theories, ΔL is designed as the whole fraction of the tube that did not survive from the full arm retraction by that time. On the other hand, the Likhtman–McLeish (LM) approach has a special scaling form for the effective tube length:(16)$$\begin{array}{c} \frac{L_{eff}}{L}=1-\frac{1.69}{\sqrt{Z}}+\frac{2}{Z}-\frac{1.24}{Z^{3/2}}. \end{array}$$

Introducing CLF into reptation allowed explaining the experimentally observed dependency $\tau_{d}∼M^{3.4}\;$[9]. Doi suggested [12] that the effective length, 1−ΔL/L, decreases with the chain length as $1-X/\sqrt{Z}$ and *X* being a constant larger than 1.47. Stephanou predicted [67] the value of *X* from the primitive path analysis to be close to $1/\sqrt{3}$.

In Figure 4 we plot the estimations of the effective fraction of the tube 1−ΔL/L participating in reptation versus *Z* for polymers with a molecular weight up to 400 entanglements. As expected, the Likhtman–McLeish results are close to Doi’s, where the chain ends move quickly on the time scale of the Rouse time $\tau_{R}$. On the other hand, the primitive path (PP) simulations reveal a much weaker effect of CLF. Similar observations were done by Likhtman in his MD simulations of polymer melts [68]: chains with the length of *Z* = 7–15 had tube shortening by only 13%, instead of 32–42%, that we determined here, based on the BoB model. Conversely, according to the predictions of BoB, CLF has a much stronger impact on reptation than it was assumed initially. Prior to giving any details, we note that the BoB model correctly estimates that part of the chain which relaxes by retraction rather than by reptation (based on survival probabilities). However, subtracting this length from the length to be relaxed by reptation is, to our opinion, a too strong assumption.

At the other extreme, TMA, by excluding completely the influence of the fluctuation process on the reptation time, misses a quite important fundamental concept in its modeling. In order to include this effect, we need first to highlight an important point: only the segments relaxing fast in comparison to the reptation should contribute to the shortening of the tube during diffusion along the tube. Therefore, one should answer the question: which portion of the linear chain is fluctuating rapidly enough to be considered negligible for reptation? Similarly to the way we defined a possible solvent effect in reptation process [8], we argue that it is estimated as the part of the tube which has relaxed due to fluctuations by time $t=\tau_{d}/Z^{2}=3Z\tau_{e}$. In the other words, a chain segment can take advantage of the widened effective tube, if its constraint release characteristic time resulting from CLF is on the same time scale as the reptation time. After calculating values of $L_{eff}$ for a wide range of *Z* we follow the approach of Likhtman [4] and conclude that the effective length of the primitive path can be well approximated by the empirical equation
(17)$$\begin{array}{c} \frac{L_{eff}}{L}=1-\frac{0.46}{\sqrt{Z}}+\frac{0.3}{Z}-\frac{1.86}{Z^{3/2}}. \end{array}$$

In Figure 4, the corresponding results appear as TMA2 in the new coordinate system and is very similar to the results obtained by Stephanou [67]. Based on this result, a new reptation time in the new coordinate system can be defined as $\tau_{d}^{new}=\tau_{d}^{0}{\left(1-x_{0}\right)}^{3}{\left(L_{eff}/L\right)}^{2}$, where $\tau_{d}^{0}$ is the initial reptation time calculated in the old coordinate system according to Equation (14). In order to be consistent in a comparison of the part relaxed by CLF with the ones found with the other models, we can account for the amount of the material relaxed before $\tau_{e}$ , and use the old coordinate system. In such a case, the tube is predicted to be shortened by CLF (TMA1 in Figure 4), as
(18)$$\begin{array}{c} \frac{L_{eff}^{*}}{L}=\max\left(0.6,1-\frac{0.17}{\sqrt{Z}}-\frac{2.98}{Z}+\frac{1.9}{Z^{3/2}}\right), \end{array}$$
with the reptation time $\tau_{d}^{new}=\tau_{d}^{0}\left(1-x_{0}\right){\left(L_{eff}^{*}/L\right)}^{2}$. The lower limit 0.6 for CLF influence is obtained empirically and plays a role only for very short chains. The results obtained by using this formula in TMA can be seen in Figure 2, where PBD experimental data are compared to the corresponding predictions. Now TMA properly captures the behavior of linear polymers. Note that the effect of CLF on reptation time is important only for polymers with molecular weight up to 50 entanglements. For higher molecular weights it becomes negligible.

## 6. Comparison between the TMA and BoB Models

In this section, we discuss important differences between the TMA and BoB models. This includes designing the expression used for describing the relaxation modulus, the influence of retraction process, as well as the way reptation and CLF are interconnected.

### 6.1. Relaxation Modulus

At first, we intend to show how the stress relaxation formula used in the BoB and the Hierarchical models is derived from the continuous Maxwell model. This allows us, in a second step, to show how the CR process is taken into account in this expression. The relaxation function for the viscoelastic fluid
(19)$$\begin{array}{c} G\left(t\right)=\int_{0}^{+\infty }H\left(\tau \right)\exp\left(-t/\tau \right)\;d\tau \end{array}$$
is described by a continuous distribution of Maxwell modes, called the relaxation spectrum H(τ).

Rheological relaxation processes tend to occur over so long periods of time that relaxation processes are usually modeled using logarithmic time. Nevertheless, in the derivations further on we use time on a linear scale. The exponential function is called the kernel. The relaxation spectrum H(τ) is independent of time and is the fundamental description of the system. It can be calculated from the Alfrey approximation [69], where a set of exponents is replaced by a set of step functions. Simply, exp(−t/τ)≈0, at time t>τ and exp(−t/τ)≈1, at time t<τ. Equation (19) is then rewritten as
(20)$$\begin{array}{c} G\left(t\right)\approx \int_{t}^{+\infty }H\left(\tau \right)\;d\tau , \end{array}$$
so that
(21)$$\begin{array}{c} H\left(\tau \right)\approx -{\left[\frac{dG(t)}{dt}\right]}_{\tau }. \end{array}$$

It should be highlighted that the integral in Equation (19), with the relaxation spectrum from Equation (21), represents the stress decay through time and is always defined in the interval [0,+∞]. This means that a single Maxwell element is defined as an amount of the stress decrease at relaxation time τ. Then H(τ)dτ defines the fraction of Maxwell’s elements with relaxation times between τ and τ+dτ, namely, an amount of stress relaxed within that time interval.

Using Equation (21) with $G\left(t\right)=G_{N}^{0}\Psi \left(t\right)\Phi^{\alpha }\left(t\right)$, together with Equation (19), the BoB and the Hierarchical models, define the relaxation modulus as:(22)$$\begin{array}{cc} \frac{G(t)}{G_{N}^{0}} & =\int_{\tau_{start}}^{\tau_{stop}}{\left[-\frac{d[\Psi \left(t\right)\Phi^{\alpha }\left(t\right)]}{dt}\right]}_{\tau }\exp\left(-t/\tau \right)\;d\tau , \end{array}$$
where Ψ(τ) is the amount of unrelaxed material at time τ and Φ(τ) is the volume fraction which corresponds to the tube diameter (“supertube” fraction). Both models start from $\tau_{start}=0$ and at each computational step, they increase the time by dτ, up to $\tau_{stop}=\tau_{d}+d\tau$. Here, $\tau_{d}=m\;d\tau$ is the reptation time, and m+1 is the total number of steps. This means that although the simulation time is longer than $\tau_{stop}$, the calculation of Ψ(t) and Φ(t) stops after m+1 steps. Thus, this model step is based on the assumption that reptation happens instantaneously at $\tau =\tau_{d}$ and Ψ(t) becomes zero at the last computational step. Later, we show that this approximation leads to an overestimation of the reptation time.

Since the modeling of BoB is applied in the discrete time domain, all variables are calculated at each time step. The value of the integral in Equation (22) is approximated numerically by computing the product of the value of the integrand at the end-point of the interval by the interval length. Given the values at each time step we can approximate the integrand using a linear spline and α=1 [7]. Applying the numerical approximation of the derivative product rule, the integral over the interval $\left[\tau_{n},\tau_{n+1}\right]$ becomes
(23)$$\begin{array}{c} \int_{\tau_{n}}^{\tau_{n+1}}{\left[-\frac{d[\Psi \left(t\right)\Phi^{\alpha }\left(t\right)]}{dt}\right]}_{\tau }\exp\left(-t/\tau \right)\;d\tau =F\left(\tau_{n+1}\right)\exp\left(-t/\tau_{n+1}\right), \end{array}$$
where
(24)$$\begin{array}{c} F\left(\tau_{n+1}\right)=\Phi \left(\tau_{n+1}\right)\left(\Psi \left(\tau_{n}\right)-\Psi \left(\tau_{n+1}\right)\right)+\Psi \left(\tau_{n+1}\right)\left(\Phi \left(\tau_{n}\right)-\Phi \left(\tau_{n+1}\right)\right). \end{array}$$

Thus, the numerical approximation of Equation (22) for the BoB model becomes
(25)$$\begin{array}{cc} \frac{G(t)}{G_{N}^{0}} & =\sum_{k=1}^{m+1}F\left(\tau_{k}\right)\exp\left(-t/\tau_{k}\right). \end{array}$$

In order to separate contributions of retraction and reptation in the relaxation modulus, we rewrite Equation (22) for the time interval [0,+∞] such that only the reptation integrand from $\tau_{d}$ till +∞ is numerically approximated
(26)$$\begin{array}{ll} \frac{G(t)}{G_{N}^{0}} & =\int_{0}^{\tau_{d}}{\left[-\frac{d[\Psi (t)\Phi (t)]}{dt}\right]}_{\tau }\exp\left(-t/\tau \right)\;d\tau +\int_{\tau_{d}}^{+\infty }{\left[-\frac{d[\Psi (t)\Phi (t)]}{dt}\right]}_{\tau }\exp\left(-t/\tau \right)\;d\tau \\ & \approx \int_{0}^{\tau_{d}}{\left[-\frac{d[\Psi (t)\Phi (t)]}{dt}\right]}_{\tau }\exp\left(-t/\tau \right)\;d\tau +\sum_{n=1}^{+\infty }F\left(\tau_{d}+nd\tau \right)\exp\left(-t/\left(\tau_{d}+nd\tau \right)\right)\\ & =\int_{0}^{\tau_{d}}{\left[-\frac{d[\Psi (t)\Phi (t)]}{dt}\right]}_{\tau }\exp\left(-t/\tau \right)\;d\tau +\Phi \left(\tau_{d}+d\tau \right)\Psi \left(\tau_{d}\right)\exp(-t/\left(\tau_{d}+nd\tau \right)\\ & \approx G_{retr}\left(t\right)+G\left(\tau_{d}\right)\exp\left(-t/\tau_{d}\right)\\ & =G_{retr}\left(t\right)+G_{rept}\left(t\right). \end{array}$$

The reptation integrand has been simplified due to the above mentioned assumption, which leads to $\Psi (\tau_{d}+nd\tau )=0$ for n∈[2,+∞].

The resulting Equation (26) is nothing more than the expression for the relaxation modulus in the Milner–McLeish model. The first term is a portion of the stress relaxing by arm retraction and has a form identical to the one used in other models, including the TMA. However, the second term which is responsible for the reptation differs from the one in TMA. In the Milner–McLeish theory, this has the following form
(27)$$\begin{array}{c} G_{rept}\left(t\right)=G\left(\tau_{d}\right)\sum_{p_{odd}}\left(8/\pi^{2}\right)p^{-2}\exp\left(-p^{2}t/\tau_{d}\right)≃G\left(\tau_{d}\right)\exp\left(-t/\tau_{d}\right), \end{array}$$
where $G(\tau_{d})$ is the fraction of the stress which remains after the relaxation by retraction and is identical to $\Psi \left(\tau_{d}\right)\Phi \left(\tau_{d}\right)$ in Equation (26). The sum over weighted relaxation times in Equation (27) is heavily dominated by the longest relaxation time $\tau_{d}$, and thus, can be simplified. Looking at Equation (27), we see that it excludes the contribution of constraint release to reptation, as it is explicitly stated in Milner–McLeish paper [3]. That is why the exponent has the form $\exp(-t/\tau_{d})$, instead of $\exp(-2t/\tau_{d})$, when constraint release is considered. While, the BoB and the Hierarchical models assume constraint release always to be active, as suggested in Equation (22), they remove its effect in the derivation of Equation (26) and use $\exp(-t/\tau_{d})$ for the reptation part .

In fact, the idea of reptation being an instantaneous process, i.e., $\Psi (\tau_{d}+nd\tau )=0$ for n∈[2,+∞], appears to be a strong approximation. Indeed, at time $\tau_{d}$, the stress of the segments relaxing by reptation only drops to 1/e of its initial value, meaning that $\Psi (\tau_{d}+nd\tau )$ should not be 0. Since no CCR regime is considered for monodisperse chains, $\Psi (\tau_{d}+nd\tau )=0$ also means that $\Phi (\tau_{d}+nd\tau )=0$. Thus, Equation (19) with the relaxation spectrum from Equation (21) can not be used, as the maximum relaxation time of the chosen Maxwell elements should not be limited by $\tau_{d}$ and should be defined over the time interval [0,+∞]. If instead of the integral form for the relaxation modulus, one just uses the equivalent form $G\left(t\right)=G_{N}^{0}\Psi \left(t\right)\Phi^{\alpha }\left(t\right)$, then the idea of $\Psi (\tau_{d}+nd\tau )=0$, simply leads to G(t)=0, at $t>\tau_{d}+d\tau$. The latter conclusion is incorrect, as stress relaxes exponentially over a long period of time.

In order to resolve this issue, the approximation $\Psi (\tau_{d})=0$ can be used in a slightly different context. With the introduction of constraint release to the tube model, it is difficult to determine Maxwell elements for Equation (20), such that the maximum relaxation time is $\tau_{d}$. Instead, we define $\Psi^{*}\left(t\right)$ being the amount of material with relaxation time larger than *t*. Then, $\Psi^{*}\left(t\right)=\Psi \left(t\right),\;\;t\leq \tau_{d}$ and $\Psi^{*}\left(t\right)=0,\;\;t>\tau_{d}$. The resulting expression for the relaxation modulus becomes
(28)$$\begin{array}{ll} \frac{G(t)}{G_{N}^{0}} & =\Phi \left(t\right)\int_{0}^{+\infty }{\left[-\frac{d[\Psi^{*}\left(t\right)]}{dt}\right]}_{\tau }\exp\left(-t/\tau \right)\;d\tau \\ & =G_{retr}\left(t\right)+\Phi \left(t\right)\Psi \left(\tau_{d}\right)\exp\left(-t/\tau_{d}\right), \end{array}$$
where the second term represents relaxation via reptation. Recall that $\Phi \left(t>\tau_{d}\right)=\Psi \left(\tau_{d}\right)\exp\left(-t/\tau_{d}\right)$, if CCR is inactive and consequently,
(29)$$\begin{array}{c} G\left(t\right)=G_{N}^{0}\left(G_{retr}\left(t\right)+\Psi^{2}\left(\tau_{d}\right)\exp\left(-2t/\tau_{d}\right)\right). \end{array}$$

This form of relaxation modulus includes the double reptation expression, where the unrelaxed portion of the entanglements contributes to the modulus in proportion to the square of the remaining fraction of entanglements.

Keeping Equation (29) aside, we can now show that similar expression can be drawn from Equations (19) and (21) assuming that a part of the tube $\Psi_{fluc}$ is relaxed mostly by retraction up to time $\tau^{*}$ and the remaining part $\Psi_{rept}$ by reptation with the survival function given by $\Psi_{rept}\left(t\right)=\Psi_{rept}\exp\left(-t/\tau_{d}\right)$. Knowing that $\Psi_{fluc}+\Psi_{rept}=1$ and neglecting CCR (Φ(t)=Ψ(t)), Equation (22) becomes
(30)$$\begin{array}{cc} \frac{G(t)}{G_{N}^{0}} & =\int_{0}^{\tau^{*}}{\left[-\frac{d[\Psi_{fluc}^{2}\left(t\right)]}{dt}\right]}_{\tau }\exp\left(-t/\tau \right)d\tau +\int_{\tau^{*}}^{\infty }{\left[-\frac{d[\Psi_{rept}^{2}\left(t\right)]}{dt}\right]}_{\tau }\exp\left(-t/\tau \right)d\tau . \end{array}$$
Note that $\Psi_{rept}\left(t\right)$ is constant for $t<\tau^{*}$ and $d\Psi_{rept}\left(t\right)/dt=0$ in that interval. Thus, although the second integral is responsible for relaxation via reptation for $t>\tau^{*}$, integration can start from 0 rather than from $\tau^{*}$. The second term of Equation (30), can then be simplified to $\Psi_{rept}^{2}\left(t\right)=\Psi_{rept}^{2}\left(\tau_{d}\right)\exp\left(-2t/\tau_{d}\right)$, if we replace the exponential with a step function. Similarly to the G(t) derived above, this expression reproduces the concept of double reptation. In reality, reptation should start earlier than $\tau_{d}$, when $\Psi_{rept}\left(t\right)≃1$. Empirical estimations show that $\tau^{*}=\tau_{d}/\sqrt{Z}$ and the segment with this relaxation time is close to $x_{d}$, as calculated by the BoB model.

Based on this analysis, our present concern is that the BoB model fails to properly capture reptation behaviour. Due to the strong assumption, i.e., $\Psi (\tau_{d}+d\tau )=0$, the term $G\left(\tau_{d}\right)\exp\left(-t/\tau_{d}\right)$ in Equation (26) misses a factor of 2 in the exponent compared to Equation (29) (simplified version), and consequently, the relaxation time in the BoB and the Hierarchical models is smaller than the original $\tau_{d}$ by approximately a factor of 2.

Interestingly, the too slow reptation term seems to be well compensated by the too strong effect of CLF (see Figure 4). There seems however to be no justification regarding the hypothesis that deep fluctuations of chain ends speed up the reptation process.

### 6.2. Arm Retraction

If we deal with the linear chains as two-arm stars, then the contour-length fluctuations of the ends are the same as the retracting motion of a star arm. The retraction dynamics of the TMA and the BoB models is already compared in our previous work [25], where it was shown that both models successfully predict the linear viscoelastic data of star-like molecules. However, the difference in the definition of the prefactor in the first-passage time of the activated fluctuation $\tau_{fluc}\left(x\right)$, $x>x_{tr}$ together with some small algorithmic discrepancies lead to slightly different material parameters for the two models ($M_{e}^{BoB}>M_{e}^{TMA}$). Here, the best fit to the experimental data was obtained with the same entanglement spacing parameter $M_{e}$ for the TMA and BoB predictions (see Table 4). It means that the deep fluctuations of linear chains in the TMA model are slightly faster than in the BoB model. Due to that fact, we expect to see a slower relaxation of shorter linear polymers (Z<15).

### 6.3. Cumulative Effect of Reptation and CLF

All models, except TMA, assume that at some time, retraction of the linear polymer will be interrupted, and stress relaxation is completed by reptation using the theoretical framework developed by Doi and Edwards [2]. Similarly, the Hierarchical and the BoB models consider retraction and reptation as mutually exclusive processes with a transition around reptation time $\tau_{d}^{BoB}$. On the other hand, the TMA model assumes both processes to take place simultaneously and only the long chains show this obvious crossover behaviour. At first sight, it seems that these choices are the cause of the alternative definitions of reptation time in each model. However, we showed in the previous section that the reason is different. In the present work, TMA shortens the contour length by excluding the part of the chain that has relaxed by $t=3Z\tau_{e}$, whereas other models exclude much more—everything that has relaxed by $t=\tau_{d}^{BoB}$ (see Figure 4).

Naturally, it is interesting to check the viscoelastic behavior when the long chains (Z>50) have this implicit transition between reptation and retraction in the TMA model. At each time step we calculate the probabilities $p_{fluc}\left(x,t\right)$ and $p_{rep}\left(x,t\right)$ that the segment *x* survives from relaxation by fluctuation and by reptation. Furthermore, at any time *t* we can determine the so called relaxation front, a segment $x_{t}$, which separates relaxed and unrelaxed segments. This segment has a relaxation time $\tau (x_{t})=t$ and the fraction of material not yet relaxed by reptation or retraction can be found as $\Psi \left(t\right)=1-x_{t}$. Once the reptation time $\tau_{d}^{TMA}$ is reached, the relaxation front is at segment $x_{d}$. In Figure 5 and Figure 6, we plot the relaxation probabilities $p_{rept}\left(x,\tau \left(x\right)\right)$ and $p_{fluct}\left(x,\tau \left(x\right)\right)$ for segments $x<x_{d}$ at the time τ(x) they relaxed, and $p_{rept}\left(x,\tau_{d}^{TMA}\right)$ and $p_{fluct}\left(x,\tau_{d}^{TMA}\right),$ for segments $x>x_{d}$ at the reptation time $\tau_{d}^{TMA}$ estimated by Equation (14). Thus, the observed decrease of $p_{rept}\left(x,\tau \left(x\right)\right)$ with the depth of the segment, *x*, is due to the fact that for $x<x_{d}$, the time at which this probability is determined increases with *x*. By comparing $p_{rept}\left(x,\tau \left(x\right)\right)$ to $p_{fluct}\left(x,\tau \left(x\right)\right)$ one can notice that both processes cross over around segment $x_{s}$. This indicates that it is more probable that segments $x<x_{s}$ are relaxed by fluctuations rather than by reptation, and segments $x>x_{s}$ are relaxed primarily via reptation, as $p_{fluct}\left(x,\tau \left(x\right)\right)≃1$. This estimation of transition between relaxation process is identical to the one from the BoB model (see Figure 4). Fluctuations stop at segment $x_{s}$ with relaxation time $\tau_{fluc}\left(x_{s}\right)$ and reptation takes over for segments $x>x_{s}$.

If we analyse the short chains (see Figure 7), there is no clear separation between fluctuations and reptation because the probability of being relaxed by retraction <1 for all segments. However, for $x<x_{s}$ the chance to be relaxed by reptation is very low, same as for the long chains. The rest of the segments are relaxed via reptation and retraction, as $p_{rept}\left(x,t\right)≃p_{fluct}\left(x,t\right)$.

In fact, it is easy to show the numerical equivalence of the TMA and the BoB models, especially for long chains (Z>50). To illustrate this conclusion, we add several features from the BoB model to the TMA model and compare the results. First, we modify the TMA algorithm (TMA${}_{excl}$) in order to allow reptation to start only after the arm fluctuation is finished. This transition from fluctuation process to reptation takes place at segment $x_{s}$. In other words, the joint survival probability becomes $p\left(x,t\right)=p_{fluc}\left(x,t\right)$, if $x<x_{s}$ and $p\left(x,t\right)=p_{rept}\left(x,t\right)≃exp\left(-t/\tau_{d}^{TMA}\right)$, if $x>x_{s}$, where $\tau_{fluc}\left(x_{s},t\right)=\tau_{d}^{BoB}$. It must be noted that G(t) in the TMA is calculated at each time step, so we need to determine x(s) a priori. The TMA model which accounts for this first modification is called TMA${}_{excl}$. In addition, we examine how the choice of the relaxation modulus equation used in the BoB model influences the relaxation process. Therefore, we introduce one extra modification in our model, called TMA${}_{excl-BoB}$, where the reptation time $\tau_{d}^{BoB}$ and G(t) of TMA${}_{excl}$ are now determined according to BoB’s formulations, in addition to preventing fluctuations and reptation process to take place simultaneously. This version of the TMA model is almost identical to the BoB model, except for the transition between the early and the deep fluctuations and calculation of the relaxation time for the latter one, as was explained before.

Figure 8 shows a comparison between TMA (solid line), TMA${}_{excl}$ (dotted line) and TMA${}_{excl-BoB}$ (dashed line) predictions for Auhl’s PI samples from Table 2. Both TMA and TMA${}_{excl}$ models predict nearly the same results for samples with high number of entanglements. The explanation goes as follows: by using the joint survival probability $p_{rept}\left(x,t\right)p_{fluc}\left(x,t\right)$, TMA introduces a smooth transition between retraction and reptation processes. The domination of the arm retraction ($p_{rept}\left(x,t\right)≃1$ for all unrelaxed segments *x*) is followed by a period when reptation starts to take over; namely $p_{rept}\left(x,t\right)$ tends to $p_{fluc}\left(x,t\right)$ for segments of the relaxation front. Finally, a chain will complete its relaxation by reptation, rather than by retraction, since $p_{rept}\left(x,t\right)\ll p_{fluc}\left(x,t\right)≃1$ for unrelaxed segments. In general, this rule holds for linear chains with a molecular weight of 50 entanglements or higher.

A different behavior is observed for shorter chains, where TMA${}_{excl}$ shows a slower decay of $G^{\prime }$ and $G^{\prime \prime }$ in the terminal zone. Although TMA${}_{excl}$ and TMA have the same reptation time, there is no actual transition between processes in the TMA model, as in case of the long chains. Relaxation of segments after $x_{s}$ is also governed by those two processes, rather than by pure reptation, because probabilities of being relaxed by reptation or retraction are almost equivalent for those segments. This explains the difference in the obtained results of the two models, as in TMA${}_{excl}$ relaxation ends with pure reptation.

Interestingly, comparing these data with the TMA${}_{excl-BoB}$ model allows us to illustrate how the two main approximations discussed above, i.e., a reptation process two times slower and an overestimated influence of CLF on reptation, compensate each other in the BoB model. Indeed, BoB can be viewed as the TMA${}_{excl}$ model with the reptation time $\tau_{d}^{TMA}\approx 2\tau_{d}^{BoB}$. Nevertheless, samples having a high molecular weight show slightly slower terminal relaxation with BoB. Indeed, when $\tau_{d}^{TMA}/\tau_{d}^{BoB}>2$ for short chains, TMA still has retraction, which implicitly reduces reptation time to the level of $\tau_{d}^{TMA}=2\tau_{d}^{BoB}$. For very long chains the reptation relaxation time ratio is slightly less than 2 and leads to longer terminal relaxation in the BoB model.

In the BoB model, the missing constraint release in reptation is compensated by choosing a smaller value for $\tau_{d}$ by excluding the part of the chain relaxed by deep fluctuations, such that a relationship between TMA and BoB reptation times of $\tau_{d}^{TMA}\approx 2\tau_{d}^{BoB}$ is found. Here, we argue against this hypothesis. Indeed, during retraction, one end of the chain will be first dragged inside the tube over a maximum distance $\Delta L_{ret}$ by forming some big loops between entanglements. However, at the same time, the other end will be pulled out of the tube by reptation over the same distance $\Delta L_{ret}$ and will first stretch those loops rather than translating them over the distance $\Delta L_{ret}$, so that the fluctuating tip of the chain will stay fixed. Therefore, on average, the equilibrium length of the chain stays constant. Only the fast fluctuations (compared to the reptation time) should speed up reptation. Furthermore, deep retraction cannot be considered as a fast process compared to reptation.

## 7. Comparison of Theoretical and Experimental LVE Data

First, we summarize in Table 5 the important features of the TMA, the modified TMA and the Bob which were discussed in this paper.

The modified TMA and the BoB models with the input parameters presented in Table 4 are used to compute G(ω). The comparison between experimental and theoretical data for PI samples is shown in Figure 9 and Figure 10.

The overall agreement between the Auhl’s PI data and the more advanced version of the TMA model is remarkable. The BoB model exhibits a deviation from the experimental plateau modulus for the short chains. This is explained by accounting twice the effect of chain ends relaxation as discussed in Section 4. Also, the predictions of BoB for short chains (Z<20) are relaxing a bit slower than the ones from TMA. We identified the nature of this discrepancy previously: the largest part of the chain is relaxed by retraction, and BoB would need larger $M_{e}$ to coincide with TMA predictions [25]. For middle range molecular weight samples, the TMA and the BoB models are found to yield similar results when using the same set of input parameters.

Now we consider the experimental data available for polybutadiene. Figure 11, Figure 12, Figure 13, Figure 14 and Figure 15 shows the comparison of predictions made by the TMA and the BoB models for the data from Table 1.

All the PBD data are predicted with the same values for the input parameters $M_{e}$ and $\tau_{e}$. Recall that Likhtman and McLeish [4] had to set the corresponding values differently for Baumgaertel’s [37] and Lee’s [38] data. The results in Figure 11, Figure 12, Figure 13, Figure 14 and Figure 15 show the same quality of predictions as for PI samples. All deviations between the experimental data and the BoB model are already explained above.

Finally, in Figure 16 and Figure 17 we compare our predictions for linear viscoelasticity with the experimental data of linear PS samples coming from several research groups.

Again, the predictions made by the TMA model are in a good agreement with all data, while the BoB model overpredicts the relaxation of really short chains together with a clear deviation from the plateau modulus. The data from Graessley [47] and Schausberger [46] is also considered in the work of Likhtman and McLeish [4]. To fit the data accurately, they had to adjust the values of $M_{e}$, $\tau_{e}$ and $G_{N}^{0}$, separately for each set. On the contrary, we use the same value of $M_{e}$ and $G_{N}^{0}$ for the calculations employing these samples.

There are several comments regarding the comparison of the data. First, the shape of the relaxation spectra $G^{\prime }\left(\omega \right)$ and $G^{\prime \prime }\left(\omega \right)$ predicted by TMA is in good agreement with the experimental data. Second, the agreement between TMA and BoB is quite reasonable, but with some deviations. The most pronounced disagreement is observed for the short-chain samples, where the BoB model overestimates the terminal relaxation time, and predictions are slightly shifted downwards. Presumably, this effect can be attributed to the difference in the coordinate systems used in both models, since TMA${}_{excl-BoB}$ shows a better fit. As discussed in Section 4, the value of *Z* for short chains is somewhat lower due to the renormalization. Note that the BoB model cannot predict linear and star polymers with the same $M_{e}$ value ($M_{e}^{star}>M_{e}^{linear}$). Thus, BoB predictions could lead to a better agreement for these chains, if the values of $M_{e}$ for the star-shaped polymers were used. This fix is consistent with the fact that the relaxation of the short linear polymers is mostly governed by fluctuations. Third, the distinction between BoB and TMA is expected to be more pronounced for highly entangled samples (Z>400). Nonetheless, the lack of experimental data prevent this comparison.

## 8. Conclusions

Although most of tube-based models produce similar quantitative results for the monodisperse linear viscoelastic behavior of linear macromolecules, there are some crucial differences between them; especially between TMA and the others. In this paper, we compared the theoretical frameworks for several well-known models: Milner–McLeish, Likhtman–McLeish, the Hierarchical model, the BoB model and the TMA model. Our comparison revealed that the most pronounced distinction between these models is the way the dynamic modulus G(t) is determined, and as a consequence, the relaxation time $\tau_{d}$. Additionally, the description of the transition between relaxation processes also plays an important role in the predictions, especially for shorter polymers.

Our analyses indicate that the BoB and the Hierarchical models use the same expression for the relaxation modulus as MM theory, i.e., excluding the effect of constraint release modes arising from reptation of other chains. This assumption is compensated in the MM theory, the BoB and the Hierarchical models by including the possible deep retraction of the chain extremities into CLF calculations and by removing this CLF from the reptation time. In this work, we showed that only fast fluctuations of chain ends should speed up reptation, as initially proposed in the tube theory.

On the other hand, we discussed the consequences of an incorrect approximation applied in the derivation of the relaxation modulus and proposed several alternative ways to obtain the correct expression for G(t) calculation. We also showed that, the crude assumption made in the BoB model does not dramatically affect the quality of the results for linear chains. However, one can wonder what would be their impact in the case of binary blends or for very polydisperse samples. Furthermore, modeling of branched polymers might also be affected.

Based on the comparison to experimental data, we added several modifications to the time-marching algorithm (TMA) for the case of monodisperse linear molecules. Thanks to renormalization of the coordinate system, we avoid considering the segments which relax before $\tau_{e}$ in further calculations. This procedure helped us to improve the performance of the TMA model and to eliminate the discrepancy of tube model predictions for describing the plateau modulus for short chains. Furthermore, we included the influence of the fluctuation process on reptation time in the modeling. This quite important fundamental concept was missing in the TMA model. The amount of material excluded from the reptation process is consistent with the original CLF concept and agrees with values extracted from MD simulations.

We validated the modified TMA model on a wide range of nearly monodisperse polyisoprene, polystyrene and polybutadiene linear melt samples. All predictions are in good agreement with the experimental data. Notably, same entanglement molecular weight $M_{e}$ could be used for each chemical group, contrarily to the work of Likhtman and McLeish [4], who used different values of $M_{e}$ within a chemical group. Furthermore, we were able to use the same parameters as the ones used to describe the viscoelastic properties of start polymers. Therefore, this gives us a solid base to extend tube models to polydisperse blends of linear and star polymers. Validation of the BoB model was performed on the same set of experimental data. The numerical predictions of the BoB model also fit the data quite well, apart from the deviations in the case of short chains. The latter observation can be attributed to the fact that in the BoB model, stress relaxation by contour length fluctuations is calculated starting at time t=0.

## Figures and Tables

**Figure 1 polymers-11-00754-f001:**
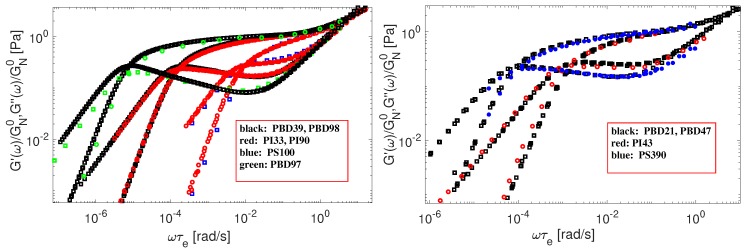
Comparison of normalized master curves between four sets of experimental data. On the left: PBD98 (□, 49.9Z) and PBD97 (□, 49Z); PBD39 (□, 19.5Z) and PI90 (∘, 19.7Z); PS100 (□, 6.7Z) and PI33 (∘, 7Z). On the right: PI43 (□, 12.1Z) and PBD22 (*, 11.5Z); PBD47 (□, 26.2Z) and PS390 (*, 25.4Z).

**Figure 2 polymers-11-00754-f002:**
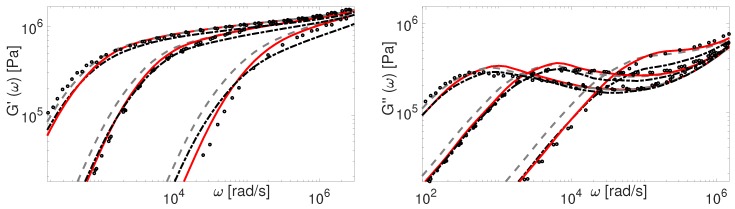
Comparison of the original TMA (━ ▪ ━), TMA in a new coordinate system (━ ━ ━) and TMA in a new coordinate system and with fast CLF subtracted (━) with experimental data: PBD8 ($4.6M_{e}$), PBD21 ($11.5M_{e}$) and PBD37 ($19.4M_{e}$).

**Figure 3 polymers-11-00754-f003:**
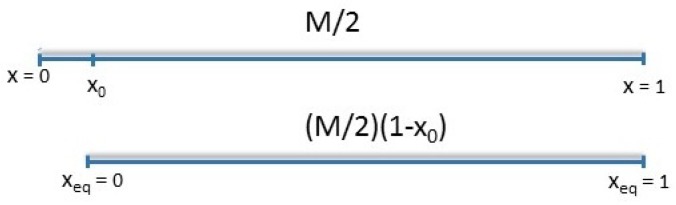
Definition of the new coordinate system. Here, M/2 is the molecular weight of the chain and $L_{0}/2$ is its equilibrium tube length in the original system of coordinates. $\left(M/2\right)\left(1-x_{0}\right)$ and $L_{eq}=L_{0}{\left(1-x_{0}\right)}^{1/2}$ are the same parameters defined in the new coordinate system.

**Figure 4 polymers-11-00754-f004:**
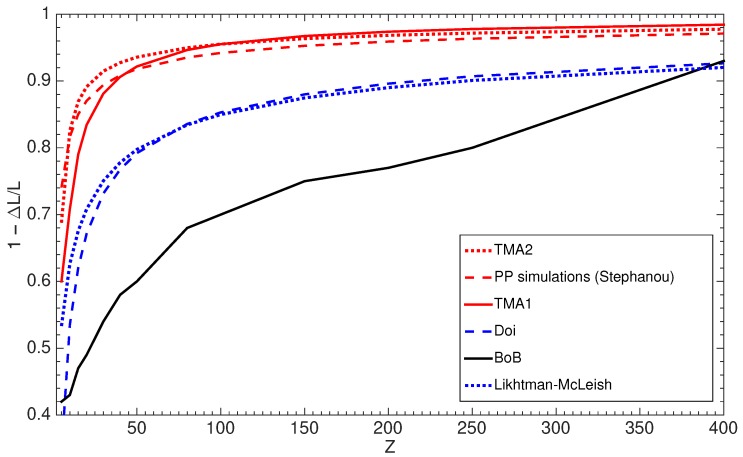
Average fluctuation of the tube contour length used in the currently existing models for determining the reptation time. TMA estimations are presented for the non-normalized (TMA1) and for normalized (TMA2) coordinate systems.

**Figure 5 polymers-11-00754-f005:**
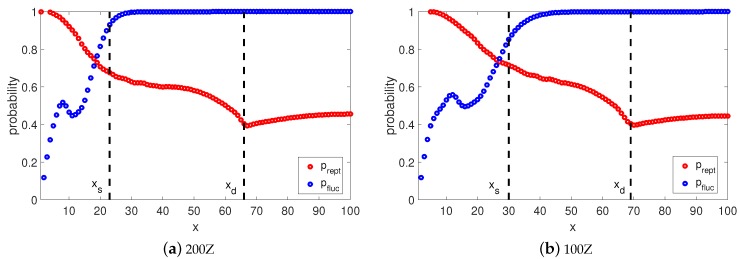
Comparison of survival probabilities $p_{fluc}\left(x,t\right)$ and $p_{rep}\left(x,t\right)$ for samples with molecular weight 200Z (**a**) and 100Z (**b**). For segments $x<x_{d}$ the probabilities are determined at the relaxation time of each segment τ(x) and for segments $x>x_{d}$ at reptation time $\tau_{d}^{TMA}$.

**Figure 6 polymers-11-00754-f006:**
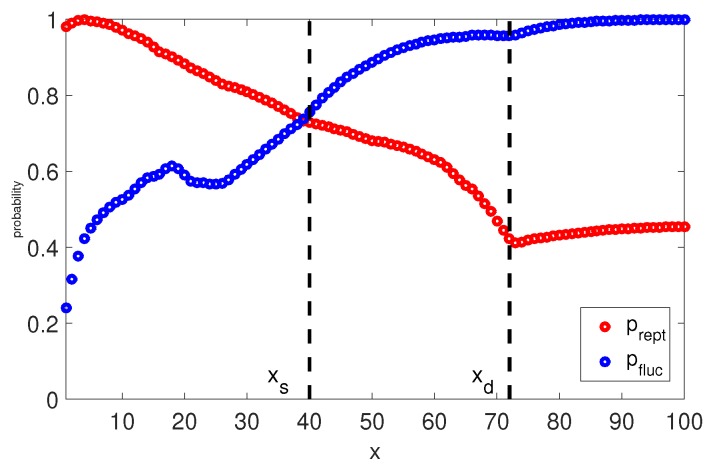
Comparison of survival probabilities $p_{fluc}\left(x,t\right)$ and $p_{rep}\left(x,t\right)$ for sample with molecular weight 47Z. For segments $x<x_{d}$ the probabilities are determined at the relaxation time of each segment τ(x) and for segments $x>x_{d}$ at reptation time $\tau_{d}^{TMA}$.

**Figure 7 polymers-11-00754-f007:**
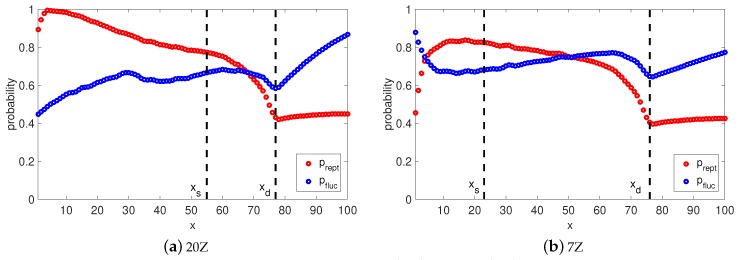
Comparison of survival probabilities $p_{fluc}\left(x,t\right)$ and $p_{rep}\left(x,t\right)$ for samples with molecular weight 20Z (**a**) and 7Z (**b**). For segments $x<x_{d}$ the probabilities are determined at the relaxation time of each segment τ(x) and for segments $x>x_{d}$ at reptation time $\tau_{d}^{TMA}$.

**Figure 8 polymers-11-00754-f008:**
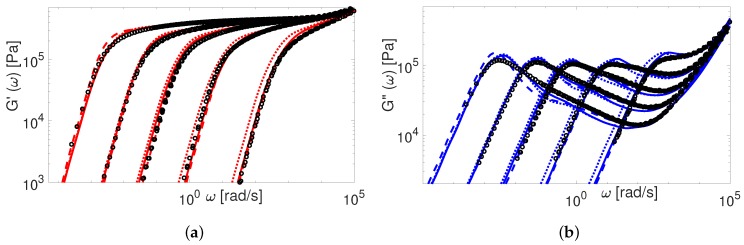
Comparison of TMA (━), TMA (▪ ▪ ▪) and TMA (━ ━ ━) predictions of $G^{\prime }$ (**a**) and $G^{\prime \prime }$ (**b**) with the experimental data (∘) for PI33 ($7M_{e}$), PI90 ($19.7M_{e}$), PI200 ($46.9M_{e}$), PI400 ($100.2M_{e}$) and PI1000 ($234.6M_{e}$).

**Figure 9 polymers-11-00754-f009:**
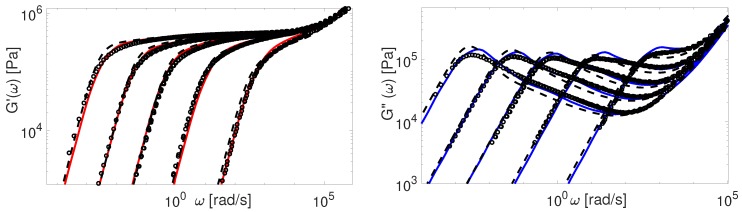
Comparison of the TMA predictions of $G^{\prime }$ (━) and $G^{\prime \prime }$ (━) with experimental data (∘) and BoB modeling (━ ━ ━) for Auhl’s PI samples.

**Figure 10 polymers-11-00754-f010:**
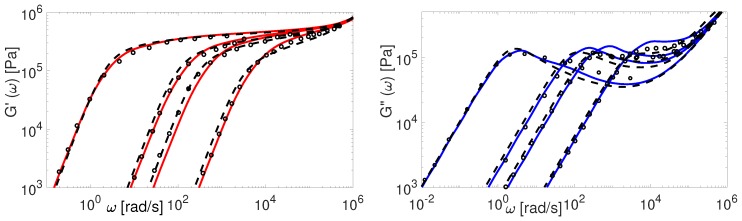
Comparison of the TMA predictions of *G*′ (━) and $G^{\prime \prime }$ (━) with experimental data (∘) and BoB modeling (━ ━ ━) for Watanabe’s PI samples.

**Figure 11 polymers-11-00754-f011:**
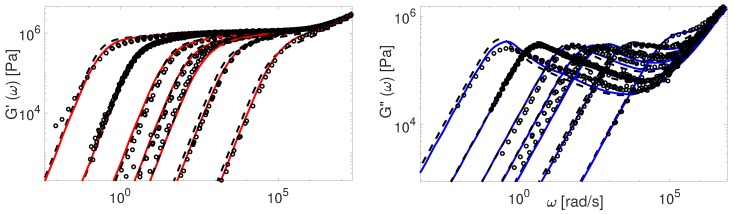
Comparison of the TMA predictions of $G^{\prime }$ (━) and $G^{\prime \prime }$ (━) with experimental data (∘) and BoB modeling (━ ━ ━) for Kapnistos’s PBD samples.

**Figure 12 polymers-11-00754-f012:**
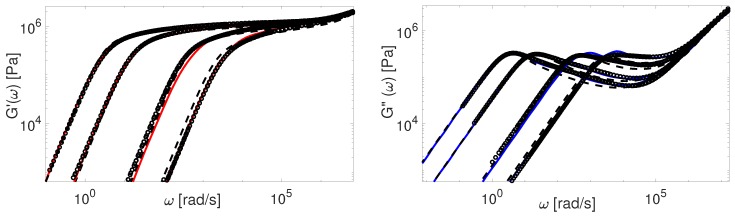
Comparison of the TMA predictions of $G^{\prime }$ (━) and $G^{\prime \prime }$ (━) with experimental data (∘) and BoB modeling (━ ━ ━) for PBD Liu’s samples.

**Figure 13 polymers-11-00754-f013:**
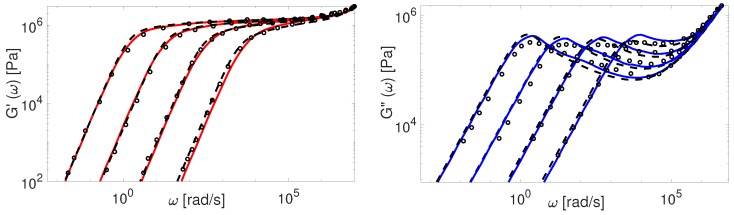
Comparison of the TMA predictions of $G^{\prime }$ (━) and $G^{\prime \prime }$ (━) with experimental data (∘) and BoB modeling (━ ━ ━) for PBD Baumgaertel’s samples.

**Figure 14 polymers-11-00754-f014:**
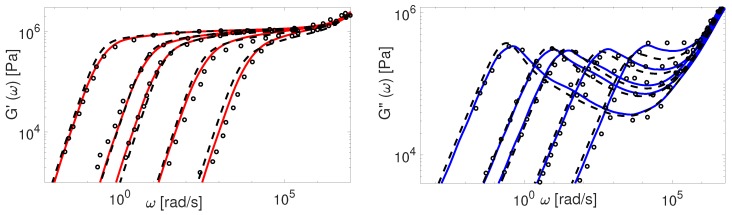
Comparison of the TMA predictions of $G^{\prime }$ (━) and $G^{\prime \prime }$ (━) with experimental data (∘) and BoB modeling (━ ━ ━) for PBD Juliani’s samples.

**Figure 15 polymers-11-00754-f015:**
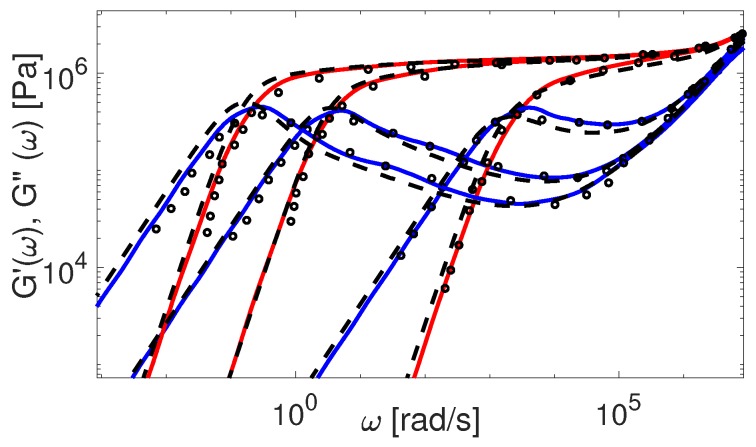
Comparison of the TMA predictions of $G^{\prime }$ (━) and $G^{\prime \prime }$ (━) with experimental data (∘) and BoB modeling (━ ━ ━) for PBD Lee’s samples.

**Figure 16 polymers-11-00754-f016:**
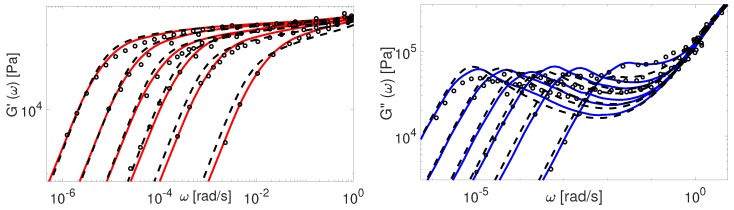
Comparison of the TMA predictions of $G^{\prime }$ (━) and $G^{\prime \prime }$ (━) with experimental data (∘) and BoB modeling (━ ━ ━) for PS Nielsen, Huang and Shahid samples.

**Figure 17 polymers-11-00754-f017:**
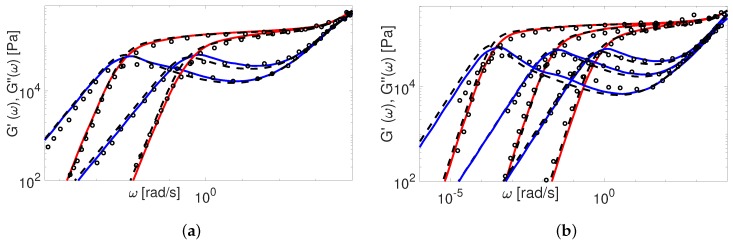
Comparison of the TMA predictions of $G^{\prime }$ (━) and $G^{\prime \prime }$ (━) with experimental data (∘) and BoB modeling (━ ━ ━) for PS (**a**) Graessley and (**b**) Schausberger samples.

**Table 1 polymers-11-00754-t001:** Characteristics of linear monodisperse PBD samples.

Sample	$M_{w}$ (g/mol)	*Z*	PI	T °C
**Kapnistos**
PBD8	9100	4.6	1.04	25.0
PBD21	22,800	11.5	1.05	25.0
PBD37	38,400	19.4	1.05	25.0
PBD47	51,800	26.2	1.11	25.0
PBD75	78,800	39.8	1.05	25.0
PBD165	155,000	78.3	1.05	25.0
PBD326	365,000	184.3	1.12	25.0
**Liu**
PBD22	22,800	11.5	1.05	25.0
PBD39	38,600	19.5	1.03	25.0
PBD98	98,800	49.9	1.03	25.0
PBD163	163,000	82.3	1.01	25.0
**Baumgaertel**
PBD20	20,700	10.5	1.04	28.0
PBD44	44,100	22.3	1.04	28.0
PBD97	97,000	49.0	1.07	28.0
PBD201	201,000	101.5	1.27	28.0
**Lee**
PBD25	26,060	13.2	1.01	28.0
PBD161	163,220	82.4	1.01	28.0
PBD395	402,900	203.5	1.02	28.0
**Juliani**
PBD18	18,540	9.3	1.03	26.0
PBD41	42,230	21.3	1.03	26.0
PBD86	89,960	45.4	1.04	26.0
PBD129	133,180	67.2	1.03	26.0
PBD336	359,520	181.6	1.07	26.0

**Table 2 polymers-11-00754-t002:** Characteristics of linear monodisperse PI samples.

Sample	$M_{w}$ (g/mol)	*Z*	PI	T °C
**Auhl**
PI33	33,600	7.0	1.03	25.0
PI90	94,900	19.7	1.03	25.0
PI200	226,000	46.9	1.03	25.0
PI400	483,000	100.2	1.03	25.0
PI1000	1,131,000	234.6	1.05	25.0
**Watanabe**
PI21	21,400	6.0	1.04	40.0
PI43	43,200	12.1	1.03	40.0
PI60	59,900	16.8	1.05	40.0
PI179	179,000	50.0	1.02	40.0

**Table 3 polymers-11-00754-t003:** Characteristics of linear monodisperse PS samples.

Sample	$M_{w}$ (g/mol)	*Z*	PI	T ${}^{\circ }C$
**Nielsen**
PS100	102,800	6.7	1.02	130.0
PS200	200,000	13.0	1.04	130.0
PS390	390,000	25.4	1.06	130.0
**Huang**
PS285	285,000	18.6	1.09	130.0
PS545	545,000	35.5	1.12	130.0
**Shahid**
PS820	820,000	53.4	1.02	130.0
**Schausberger**
PS290	290,000	18.9	1.07	180.0
PS750	750,000	48.9	1.07	180.0
PS3000	3,000,000	195.4	1.05	180.0
**Graessley**
PS275	275,000	17.9		169.5
PS860	860,000	56.0		169.5

**Table 4 polymers-11-00754-t004:** The TMA and BoB input parameters for PBD and PI.

Sample	TMA and BoB			
	$G_{N(\mathrm{used})}^{0}$ (MPa)	$G_{N(\mathrm{calc})}^{0}$ (MPa)	$M_{e}$ (g/m)	$\tau_{e}$ (s)
**PBD**				
25.0 °C	1.15	0.9	1980	$4.0\times 10^{-7}$
26.0 °C	1.10			
28.0 °C	1.45			
**PI**			
25.0 °C	0.46	0.38	4820	$1.6\times 10^{-5}$
40.0 °C	0.48	0.50	3575	$2.7\times 10^{-6}$
**PS**			
130.0 °C	0.23	0.18	15,350	$7.5\times 10^{-1}$
130.0 °C ${}^{a}$	0.21			$7.5\times 10^{-1}$
169.5 °C		0.19		$9.2\times 10^{-4}$
180.0 °C		0.19		$3.4\times 10^{-4}$

${}^{a}$ Nielsen data [44].

**Table 5 polymers-11-00754-t005:** Main differences between the TMA, the modified TMA and the BoB.

Models	CLF	Polymer Fraction Acting as Solvent for Reptation	CR Effect in Reptation	Chain Ends Treatment	Separated Relaxation Processes
**TMA**	Early fluctuations and retraction	No	Yes	No	No
**Modified TMA**	Early fluctuations and retraction	Yes	Yes	Yes	No
**BoB**	Early fluctuations and retraction	Yes	No	No	Yes

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
