# Peer review of "Comparative Analysis of Different Tube Models for Linear Rheology of Monodisperse Linear Entangled Polymers"

_polymers, 2019, doi:10.3390/polym11050754_

Round 1
Reviewer 1 Report
Review for “Polymers” Manuscript ID: polymers-457273
Title: Comparative Analysis of Different Tube Models for Linear Rheology of Linear Entangled Polymers
From the title, it seems that the authors would like to compare different models for predicting LVE of linear entangled polymers. However, this paper only tested on monodisperse polymers, where constraint-release effect is not important. This paper will be more interesting if verification with polymer samples having strong constraint release effects (such as polydisperse polymers or binary blends) are included. The comparison of the TMA model with the BOB model did not quite give a good insight as the Bob model as stated in this paper did not include constraint-release feature and it will sure fail in polydisperse polymers. The BOB model also assumed a clear cut between the region of Rouse mode and region of reptation mode, which is not commonly found in many other models.
However, an interesting point of this paper is that it tries to emphasize on good prediction of very short polymer chains (very small number of entanglements), which could not be achieved in other earlier models (such as Pattamaprom et al. 2000) without allowing Te to be a variable parameter at very low Nen. (The weakness of Pattamaprom et al. 2000 may be due to the too-fast contour-length fluctuation.) So this point worth emphasizing. Nevertheless, the write-up of this paper is quite difficult to follow and clear achievement of this paper could not be easily seen.
- In this regard, the authors should explore more literature whether there were any constitutive models addressing predictions of short polymer chains (low Nen) and their findings (if any).
- Too bad that, for this work, parameters like Gno, Me have to be adjusted and could not be obtained from previously tabulated reference books/papers. This would make further use of this model for prior predictions more challenging. Moreover, by adjusting too much, it would be more difficult to learn about some missing or inadequate expression of certain features.
Abstract line 9: This paper focused and verified only on monodispersed linear polymers. How can it investigate or verify the contribution of constraint release?
P. 4 last paragraph: Not true. Many many models also consider high frequency Rouse mode. By the way, retraction process that the authors mentioned in this paper is the same as the high frequency Rouse mode or not or is it the contour-length fluctuation process, which consider retraction of the chain ends?
P.6 line 5: The authors should mention that there were many models considering reptation and Rouse together by calculating their effects separately and combining them together in the calculation of G(t). (Assuming that the “retraction process” that the authors mentioned was the same as Rouse process.)
P.22 line 3-4: “the decay of CLF at time earlier than Te should be omitted………” Please elaborate more. What will happen if “the decay of CLF at time earlier than Te “ is not omitted. How would it affect short molecule and star-shape molecules?
P.22-23 maybe too lengthy and distracted. Is Rouse-like motion stated in this paper the same as the contour-length fluctuation equation proposed by Milner and McLeish (1997,1998)? If so, many models in the past already accepted and combined this Rouse CLF feature simultaneously with the reptation mode. The dominance of each mode at each position of the chain depended on the characteristic relaxation time (Treptation or Tfluctuation). However, the point on the first paragraph of P24 is ok.
P24 line 8: What does it mean by “slow down the effect of CLF”? Does it mean to make the chain relax faster (increase the effect of CLF) or relax slower (decrease the effect of CLF)? Please make it clear.
P.25 line 7 to the end: I could not understand whether for this model, the CLF mode for linear and star were treated the same way or differently? Please make clear.
P. 25 last paragraph: Why should Me be renormalized? I could understand that the number of entanglement maybe smaller due to the shortening of chain but I could not understand how relaxation of the chain end could affect the molecular weight between entanglement?
P.26 line 1: What is x0*
Figure 3: It should be (M/2)(1-xo) instead of M/2(1-xo).
P.26 line 4 from the bottom: Again, I can understand that by excluding the quickly relaxed chain end, the tube length used for the reptation mode is shorter but I could not understand why the Me (molecular weight between entanglement) is shorter.
P. 28 line 2: In this work, what is the definition of D?
P.29 line 3-4: who claimed tube shortening by 32-42%? And by which effect? CLF? Make it clear.
On page 28 line 2, what is D? How is it defined?
P.29 eq(17): How does the author come up with this equation?
P.37 last paragraph: should also illustrate the change in prept and pfluct as a function of position in the contour length at certain times to promote better understanding.
P.39 The sentence in line 6-8 is not clear. Some graphic would help for better understanding. (The graphic relevant to P.37 last paragraph may help.)
The description of the models in this paper is difficult to follow. Some additional diagram or graphic is needed to summarize important features of the model(s).
- Check on some misspelling, some missing words and some small grammatical errors throughout the paper.

Reviewer 2 Report
In this comprehensive work, the authors have compared the rheological predictions of several tube-based theoretical linear viscoelasticity models for linear polymers against experimental data for polybutadiene, polyisoprene, and polystyrene. They have performed detailed analyses on the relevant chain relaxation mechanisms, i.e., reputation, contour-length fluctuations, and constraint release, as described by the different models. Finally, they have made modifications to one of the models, the time-marching algorithm, to improve its prediction accuracy. I find the manuscript to be very well-written and of great value and interest to the polymer physics community. I suggest it is accepted, as is, for publication in Polymers.
Author Response
We thank the reviewers for the time and effort that they invested into the review of our manuscript. We were pleased by the positive evaluation of our study by Reviewer 2.
Round 2
Reviewer 1 Report
I am already ok with the authors' revised version.